# Leukamenin E Induces K8/18 Phosphorylation and Blocks the Assembly of Keratin Filament Networks Through ERK Activation

**DOI:** 10.3390/ijms21093164

**Published:** 2020-04-30

**Authors:** Bo Xia, Hui Zhang, Minghui Yang, Shilong Du, Jingxin Wei, Lan Ding

**Affiliations:** College of Life Science, Northwest Normal University, Lanzhou 730070, China; manman-zou@outlook.com (B.X.); zhanghui123gansu@163.com (H.Z.); yangmh0115@163.com (M.Y.); Mr.Du1912@Foxmail.com (S.D.); wjx20112020130@163.com (J.W.)

**Keywords:** leukamenin E 1, phosphorylation of keratin 2, inhibitory effect of keratin filaments assembly 3

## Abstract

Leukamenin E is a natural *ent*-kaurane diterpenoid isolated from *Isodon racemosa* (Hemsl) Hara that has been found to be a novel and potential keratin filament inhibitor, but its underlying mechanisms remain largely unknown. Here, we show that leukamenin E induces keratin filaments (KFs) depolymerization, largely independently of microfilament (MFs) and microtubules (MTs) in well-spread cells and inhibition of KFs assembly in spreading cells. These effects are accompanied by keratin phosphorylation at K8-Ser73/Ser431 and K18-Ser52 via the by extracellular signal-regulated kinases (ERK) pathway in primary liver carcinoma cells (PLC) and human umbilical vein endothelial cells (HUVECs). Moreover, leukamenin E increases soluble pK8-Ser73/Ser431, pK18-Ser52, and pan-keratin in the cytoplasmic supernatant by immunofluorescence imaging and Western blotting assay. Accordingly, leukamenin E inhibits the spreading and migration of cells. We propose that leukamenin E-induced keratin phosphorylation may interfere with the initiation of KFs assembly and block the formation of a new KFs network, leading to the inhibition of cell spreading. Leukamenin E is a potential target drug for inhibition of KFs assembly.

## 1. Introduction

Keratins are the largest subfamily of intermediate filaments (IFs) and the most abundant cellular proteins in simple epithelial cells, and they form cytoskeletal networks that contribute to cell-type-specific functions such as adhesion, migration, and metabolism [1]. KFs provide rigidity and strength while maintaining the dynamic and flexible structure of epithelial cells to cope with physical, chemical, and microbial stresses [2]. Currently, the molecular mechanisms governing keratin assembly–disassembly and their non-mechanical functions are largely unknown, in part due to the lack of useful chemical inhibitors as well as the unique properties of keratin filaments (KFs) [1,3,4]. Small-molecule probes (inhibitors or activators) have unique advantages for the analysis of cellular processes, including cytoskeleton assembly and cellular function. Cytochalasin and colchicine analogs as specific small-molecule inhibitors of microfilament (MFs) and microtubules (MTs) have become successful models for research on MFs and MTs assembly and their cellular function [5]. However, few compounds capable of inhibiting IFs assembly (for instance, KFs) have been reported to date. The structures of IFs are thought to play predominantly a more rigid, structural role, and their inherent stability, then, may be less amenable to direct disruption by small molecules [5]. Since keratins assemble rapidly and efficiently into higher-order structures *in vitro*, mechanisms must exist to disassemble keratin filaments *in vivo* during keratin network remodeling. A prime mechanism appears to be phosphorylation [6]. Phosphorylation increases keratin solubility and affects keratin network organization in simple epithelia. Keratin phosphorylation is expected to prevent the lateral alignment of nonpolar tetramers into 60 nm unit-length filaments (ULFs) and the longitudinal annealing of ULFs [7,8]. Therefore, the mechanism of keratin assembly–disassembly can be investigated by controlling the phosphorylation of keratin. Small-molecule compounds that can phosphorylate keratin to inhibit keratin assembly have similar effects as keratin inhibitors. Recent studies have shown that some small molecular compounds can phosphate keratin at specific sites, such as sphingosylphosphorylcholine (SPC) at K8-Ser431 and K18-Ser52 in Panc-1 cells, 12-Otetradecanoylphorbol-13-acetate (TPA) at K8-Ser431 in Panc-1 cells, and prostaglandins at K20-Ser13 in HT29-MTX cells [6,9,10,11].

Keratin phosphorylation is also linked to the progression of liver disease and cancer [6]. The keratin 8 and keratin 18 pair (K8/K18) is predominantly expressed in hepatocytes, pancreatic cells and, enterocytes. Their altered phosphorylation is associated with the aggregation of keratins in Mallory–Denk bodies (MDBs) found in patients with various liver diseases such as alcoholic hepatitis and alcoholic cirrhosis. MDBs comprise misfolded hyperphosphorylated K8/K18 with pK8-Ser73, pK8-Ser431, and pK18-Ser33 [6,12,13]. The K8 mutation (K8 Gly62C or G434S) leads to inhibition of adjacent phosphorylation at K8-Ser74 or K8-Ser432 in patients with liver disease [14]. K18 phosphorylation at Ser33 is related to hepatitis B virus (HBV) infection, and phosphorylation at Ser52 is a marker of liver injury [15]. In addition, perinuclear reorganization via phosphorylation of specific serine residues in keratin is involved in cell deformability, leading to increased migration of metastatic cancer cells [9,11,16,17]. SPC, TPA, and leukotriene B4 induce phosphorylation of K8-Ser431 and perinuclear reorganization of K8 filaments in Panc-1 cells while increasing migration of Panc-1 cells. Increased migratory properties have been suggested to occur as a result of reorganization [9,17,18]. However, clinicopathological analyses have led to contrary reports such as the absence or loss of phosphorylation at K8-Ser73 and K8-Ser431 being strongly correlated with tumor size, tumor stage, and lymph node metastasis in human oral squamous cell carcinoma (OSCC) and dephosphorylation at K8-Ser73 and K8-Ser431 in human colon carcinoma-derived HCT116 cells and colorectal cancer-derived DLD-1 cells resulting in acceleration of cancer cell motility, invasion, and metastasis [19,20,21]. Upregulation of the tumor-suppressor parkin in HeLa cells was found to be associated with increased phosphorylation of K8/K18 [22]. Taken together, these results reflect the prominent role of keratin phosphorylation in the regulation of cellular functions and the complex patterns of keratin phosphorylation regulation. Therefore, small-molecule compounds that phosphorylate keratin at different sites can also be developed as potential molecular probes for the study of these diseases.

The genus *Isodon* is composed of approximately 150 species that are widely distributed in Africa and Asia, and about 30 of these are used as folk herbs in China [23]. *Ent*-kaurane diterpenoids are especially abundant in their fresh leaves, and more than 500 *ent*-kaurane diterpenoids have been isolated from them [24]. So far, more than 1300 *ent*-kaurane diterpenoids have been found from different plant sources. The studies of these compounds that have been conducted have focused on their anti-cancer effects [25]. There are few reports of their effects on keratin [26]. A number of these compounds show significant cytotoxicity, and some have potential tumor treatment value [23,24,25,27]. Oridonin, which is an *ent*-kaurane diterpenoid, has been used as a cancer therapeutic agent in China. This compound selectively combines with the AML1-ETO oncoprotein in acute myeloid leukemia (AML) cells and induces apoptosis [28]. Other *ent*-kaurane diterpenoids, such as adamantine, epinodosin, and rabdosin B, induce the differentiation of acute promyelocytic leukemia cells by inhibiting their peroxidase or NADPH (nicotinamide adenine dinucleotide phosphate) xidase activity, which is similar to the effects of ATRA (all-trans retinoic acid), a leukemia differentiation therapeutic agent [29,30,31]. Recently, we developed an available whole-cell model to investigate the effects of drugs on KFs polymerization and reassembly in spreading, and well-spread cells removed MFs and MTs using cytochalasin B and colchicine. Leukamenin E has been shown to result in a KFs-disruption in HepG2 cells and NCI-H1299 cells independently of other major cytoskeletal components (i.e., MFs and MTs). However, the mechanisms by which leukamenin E interferes with processes of KFs polymerization *in vivo* have not been fully elucidated, although leukamenin E has been shown to bind to synthetic peptides via covalent bonds *in vitro* by mass spectrometry, suggesting a possible mechanism [26]. In the present study, we also observed that leukamenin E inhibited KFs assembly in primary liver carcinoma cells (PLC) and human umbilical vein endothelial cells (HUVECs), as shown in HepG2 cells and NCI-H1299 cells. Moreover, we demonstrated for the first time that leukamenin E-induced phosphorylation at K8-Ser73/431 and K18-Ser52 through extracellular signal-regulated kinases (ERK) activation was involved in increased soluble fraction of KFs and blocked the assembly of keratin filament network in PLC and HUVECs. Our results propose a molecular target and mechanism by which leukamenin E inhibits KFs assembly. Leukamenin E is a potential target drug for inhibition of KFs assembly.

## 2. Results

### 2.1. Effects of Leukamenin E on Cell Viability

Previous reports showed that cell apoptosis can be accompanied by keratin disassembly and reorganization of intermediate filaments [32]. To exclude apoptotic cells induced by leukamenin E for subsequent experiments, we examined the effects of leukamenin E at different concentrations on cell viability and apoptosis by MTT and acridine orange/ethidium bromide (AO/EB) staining, respectively. The leukamenin E showed obvious proliferation inhibition at 2.0–4.0 μM against PLC, HUVECs and Panc-1 cells when compared with the control group (Figure 1B), but the apoptosis rate and necrosis rate of leukamenin E-treated cells were less than 3.2% and 1.2%, respectively, in all groups (Figure 1C–E). Therefore, leukamenin E at 2.0–4.0 μM was selected for use in the following experiments. In addition, the death rate of HUVECs cells treated with leukamenin E at 12 µM for 12 h was under 6.8% by AO/EB double staining assay.

### 2.2. Leukamenin E Results in Inhibition of Keratin Assembly Independently of Other Major Cytoskeletal Components in PLC and HUVECs

#### 2.2.1. PLC and HUVEC Have a KFs Network Independent of Other Cytoplasmic Skeletons

In many cell types, deconstruction of some types of cytoskeleton filaments is usually accompanied by the collapse of another cytoskeleton [33,34], but in epithelial cell lines, there is a complete network of KFs that is relatively independent of other cytoplasmic skeletons [35]. In our previous research, KFs in HepG2 and NCI-H1299 cells were shown to be complete independently of other cytoplasmic skeletons [26]. Here, we tested whether PLC and HUVECs are suitable for subsequent experiments in which their KFs network is a complete network independent of MFs and MTs systems.

PLC and HUVECs were treated with two specific cytoskeleton-destroying drugs for removal of MFs or MF from cells. Optimal concentrations of cytochalasin B (MF inhibitors) and colchicine (MT inhibitors) were determined from the minimum doses capable of inducing the maximum observable effects on all investigated cell lines. Treatment with 8.4 µM cytochalasin B for 24 h or 0.33 µM colchicine for 2.5 h was found to be optimal for disassembly of MFs and MTs in spread cells. However, 4.2 μM cytochalasin B and 0.1 μM colchicine were optimal for disassembly of MFs and MTs in spreading cells.

In well-spread cells experiment, cells were seeded for 24 h in control medium, then transferred to medium containing 8.4 μM cytochalasin B for 24 h or 0.33 µM colchicine for 2.5 h. We observed significant disruption of the actin cytoskeleton in cytochalasin B-treated cells, including the stress fibers, filopodium, cytokinesis rings (causing binucleate cell) and peripheral MFs bundles (Figure 2A(b,j)), and almost all MTs in colchicine-treated cells were depolymerized (Figure 2A(d,l)); however, their KFs network remained intact (Figure 2A(f,h,n,p)). The cells were seeded for 24 h and then transferred to medium containing cytochalasin B for 21 h, after which 0.33 μM colchicine was added to the cells for an additional 2.5 h. We observed that both MFs and MTs were removed from cells treated with cytochalasin B and colchicine (Figure 2B(a,b,d,e)), but that cellular KFs network did not show significant alterations (Figure 2B(c,f)). These findings were similar to those observed in response to treatment with cytochalasin B alone or colchicine alone (Figure 2A(f,h,n,p)).

In a spreading cells experiment, after suspended cells were seeded in 4.2 μM cytochalasin B-containing medium and 0.1 μM colchicine-containing medium for spreading for 4–12 h, the assembly of both MFs and MTs in the treated cells was inhibited; however, the assembly of KFs and the spreading of cells was not significantly affected (Figure 3A(d–f,j–l)).

Taken together, these results suggest that the KFs network in PLC and HUVECs is relatively independent of MFs and MTs skeletons and may play key roles in spreading of these cells. Therefore, the two epithelial cell lines were used in subsequent experiments to further investigate the effects of leukamenin E on KFs assembly.

#### 2.2.2. Leukamenin E Destroys the Keratin Network in Well-Spread Cells and Inhibits the Assembly of KFs in Spreading Cells

After the well-spread cells were treated with 2.0 μM or 4.0 μM leukamenin E for 24 h, the number of KFs decreased significantly in the treated cells (Figure 4A(f,i,o,r)), and the KFs network in the cellular periphery was destroyed in some cells, showing aggregation into large bundles around the nucleus (Figure 4A(i,o,r)). The well-spread cells were transferred to medium containing both leukamenin E (2 μM) and cytochalasin B (8.4 μM) for 21 h, after which colchicine (0.33 μM) was added to medium for an additional 2.5 h. Immunofluorescence images showed that combination treatment with three drugs (leukamenin E, cytochalasin B, and colchicine) caused marked decreases in the number of KFs or collapse of KFs networks in PLC and HUVECs (Figure 4B(c,f)). We have observed that the KFs in control PLC cells are thick and dense, while the KFs in control HUVECs cells are relatively sparse, so we believe that the KF network of HUVECs cells may be more damaged than that of PLC cells when treated with leukamenin E at the same concentration.

Similarly, in a cell spreading experiment, suspended cells were seeded in medium with leukamenin E (2 μM) alone for spreading for 4–12 h. The cells treated with leukamenin E showed significant inhibition of KFs assemblies (Figure 5A(d–f,j–l)). The KFs in leukamenin E-treated cells were limited to the perinuclear region and accompanied by cell-spreading blocks (Figure 5A(d–f,j–l)).

Furthermore, western blotting assay showed that more keratin was detected in the supernatant and less in the pellet after leukamenin E treatment (Figure 7D), suggesting that leukamenin E increases keratin solubility, which may cause inhibition of the assembly of KFs in HUVECs.

Thus, the above-mentioned experiments confirm that leukamenin E can destroy the keratin network independently of other major cytoskeletal components in well-spread cells and inhibit the assembly of KFs in spreading cells.

### 2.3. Leukamenin E Increases Keratin Solubility by Phosphorylation of K8 and K18 and Inhibits the Assembly of KFs

Keratins are expressed specifically in epithelial cells, and K8 and K18 are the major keratins of simple epithelia [36,37]. The cytoskeletal organization and function of keratins seem to be regulated by serine phosphorylation in the head and/or tail of non-α-helical end domains [6,38]. *in vivo*, the major K8 and K18 phosphorylation sites are Ser52, Ser73, and Ser431, which have been implicated in altered polymerization or reorganization of KFs during different physiological states of a cell [6]. Thus, we investigated whether leukamenin E-induced collapse of KFs network in well-spread cells and inhibition of KFs assembly in spreading cells were caused by keratins phosphorylation.

The experimental results indicated that leukamenin E stimulated phosphorylation of K8 at Ser73 and Ser431 and K18 at Ser52 in well-spread PLC and HUVECs (Figure 6A), as well as in spreading HUVECs (Figure 6B). Leukamenin E-induced K8/K18 phosphorylation showed a dose-dependent effect (Figure 6A,B). The maximum phosphorylation was achieved after the well-spread cells were treated with 4 µM leukamenin E for 24 h (Figure 6A). K8 phosphorylation at Ser431 was detectable after the spreading cells were treated with 4 µM leukamenin E for 8 h or 8 µM for 4 h, while phosphorylation of both K8-Ser73 and K18-Ser52 was detectable after treatment with 4 µM for 12 h or 8 µM for 8 h (Figure 6B), showing that phosphorylation of K8-Ser431 occurred earlier than that of K8-Ser73 and K18- Ser52 in leukamenin E-treated cells. Based on previous reports that increases in the solubility of keratin were linked to phosphorylation [6], the above results suggest that phosphorylation of keratin at K8-Ser73, K8-Ser431, and K18-Ser 52 may be associated with leukamenin E-induced collapse of the KFs network in well-spread cells and inhibition of KFs assembly in spreading cells.

Keratin immunofluorescence double staining of HUVECs treated with or without leukamenin E was performed using K18 or K8-Ser431/Ser73 or K18-Ser52 antibody. In K18-stained control cells, the KFs networks were clearly intact and extended from the perinuclear zones to the cell periphery (Figure 7A(a,g,m)), while pK8-Ser431 antibody exhibited more immunoreactivity around perinuclear keratin and gradually decreased from the perinuclear position to the cellular periphery. Notably, the KFs network located on the cellular periphery showed very little staining by K8-Ser431 antibody, indicating an area of un-phosphorylation at K8-Ser431 in HUVECs (Figure 7A(b,c)). pK8-Ser73 and pK18-Ser52 staining of control cells was located in the thicker keratin fiber region (Figure 7A(h,n)). After leukamenin E treatment, the KFs networks were destroyed and reduced, and KFs aggregates were gathered around the nucleus in K8-stained cells (Figure 7A(d,j,p) and Figure 7B), while phosphorylated KFs, including pK8-Ser73, pK8-Ser431, and pK8-Ser52, were markedly decreased in the cell periphery region relative to control cells (Figure 7A(e,k,q) and Figure 7B). Moreover, more keratin or phosphorylated keratin (pK8-Ser431, pK8-Ser73, and pK8-Ser52) was detected in the supernatant, and slightly less keratin in the pellet (Figure 7C,D). Previous studies have confirmed that the assembly of keratin begins in the periphery of the cells [39], and phosphorylation has been linked to increased keratin solubility [6]. Thus, the above results suggest that leukamenin E-stimulated phosphorylation at the above three sites increased the solubility of KFs to prevent their assembly. In addition, leukamenin E does not alter the expression of keratin in PLC (Figure 6A). These findings indicate that the leukamenin E-induced decrease of KFs is not linked to the expression of keratin in PLC and HUVECs.

Keratin phosphorylation likely contributes to regulating the switch between the filamentous and soluble keratin pool (Figure 7A–D).

#### 2.3.1. Involvement of ERK in Leukamenin E-Induced KFs Phosphorylation of PLC and HUVECs

K8 is mainly phosphorylated at Ser 431 by extracellular signal-regulated kinases (ERKs) [38]. K8 reorganization is mediated through phosphorylation of specific serine residues in K8 by mitogen-activated protein kinases (MAPKs) [6,10,16,40]. Therefore, we determined whether ERK is involved in leukamenin E-induced phosphorylation at K8-Ser431, K8-Ser73, and K18-Ser52. The results revealed that unit-length E stimulated ERK phosphorylation (pERK) in a dose-dependent manner (Figure 8A–C), while leukamenin E-induced phosphorylation at K8-Ser431, K8-Ser73 and K18-Ser52 was suppressed in PLC and HUVECs treated with ERK inhibitors U0126 (Figure 8D,E). Furthermore, leukamenin E did not induce an increase in soluble keratin in HUVECs treated with ERK inhibitors U0126 (Figure 8F,G). These findings suggest that MAPK-ERK is not only involved in leukamenin E-induced phosphorylation at the three sites mentioned above but also in the increased soluble fraction of KFs.

#### 2.3.2. Transwell Migration and Wound Healing Assay

It has also been reported that keratin is linked to cell migration [6]. Krt6a/Krt6b-null keratinocytes exhibit an enhanced migration potential correlated with increased phosphorylation of specific tyrosine epitopes [41] and regulation of cell–matrix and cell–cell adhesion [42]. Some small molecular compounds that induce phosphorylation at K8-Ser431 and K8-Ser73 or K18-Ser52, such as sphingosylphosphorylcholine (SPC), 12-otetradecanoylphorbol-13-acetate (TPA), and leukotriene B4, can lead to perinuclear reorganization of K8/18 filaments and promote migration of Panc-1 cells in vitro [9,10,11,18,43]; therefore, we investigated the effects of leukamenin E on cell migration. However, leukamenin E induced an inhibitory effect on the migration of HUVECs and Panc-1 cells using both transwell and wound healing assay (Figure 9A,D,E). These findings suggest that leukamenin E might have additional dynamic regulation effects on the cytoskeleton.

## 3. Discussion

We previously demonstrated that leukamenin E, an *ent*-kaurane diterpenoid, could inhibit the assembly of keratin and destroy the keratin network in two typical epithelial cells, HepG2 and H1299 [26]. Here, we further confirmed that leukamenin E exerts its effects on keratin filaments independently of other major cytoskeletal components, MFs and MTs. Leukamenin E can inhibit the assembly of KFs, resulting in a inhibition of KF network reconstruction and decrease of KFs network in PLC and HUVECs (Figure 3A and Figure 4A(f,i,o,r) and Figure 4B(c,f)). Phosphorylation has been linked to increased keratin solubility and has shown to be involved in the precise dynamic regulation of assembly–disassembly of KFs [6,44]. Phosphorylation prevents the lateral alignment of keratin tetramers while also interfering with the next assembly stage [8,45,46], although a precise molecular understanding is still lacking [6]. Thus, we investigated whether leukamenin E is also involved in dynamic regulation of keratin by phosphorylation via the extracellular signaling pathway and increases of keratin solubility and if these effects lead to inhibition of KFs assemblies. In the present study, we identified stimulation of keratin phosphorylation at K8-Ser431, K8-Ser73, and K18-Ser52 by leukamenin E in PLC and HUVECs (Figure 6A) and showed that this was accompanied by increased soluble fraction of KFs and phosphorylated soluble fraction of K8 at Ser431/ Ser73 or K18 at Ser52 in spread cells and spreading cells (Figure 7A,B). We also found that pK8-Ser431, pK8-Ser73, and pK18-Ser52 staining of the KFs network was significantly reduced in leukamenin E-treated HUVECs (Figure 7A). These findings indicated that leukamenin E-induced reduction of KFs in well-spread cells and blocking of KFs network formation in spreading cells are linked to keratin phosphorylation at K8-Ser431, K8-Ser73, and K18-Ser52.

Previous reports confirm that KFs assembly is initiated at the periphery of the cell, and newly formed filaments and their maturation into an organized network take place in the context of a continuous centripetal flow with disassembly and turnover steps occurring near the nuclear envelope [39]. In untreated control cells, we observed a gradient distribution of pK8-Ser431 stain that extended from the perinuclear region to the cell periphery. Much more pK8-Ser431 staining was distributed around the perinuclear region, while it gradually reduced toward the cellular periphery, and a KFs region that was un-stained by pK8-Ser431 antibody was located at the edge of periphery (Figure 7A(b,c)). These results suggest that the region of disassembly and turnover may be located where the perinuclear KFs were stained by pK8-Ser431 antibody, while the newly formed filaments may be in the cellular periphery that was stained by K8, but not pK8-Ser431 antibody. After the HUVECs were treated with leukamenin E, the area of the K8 network decreased significantly (Figure 7A(d,j,p), while pK8-Ser431 stain was obviously attenuated (Figure 7A(e)). These findings indicated that increased soluble pK8-Ser431 fraction induced by leukamenin E may interfere with initiation of KFs assembly at the periphery of the cell and block the formation of a new KFs network and then inhibit turnover of KFs in HUVECs. Phosphorylation staining of K8-Ser73 and K18-Ser52 was distributed in thicker K8/18 KFs in the control cells, which differed from that of phosphorylation stain of K8-Ser431 (Figure 7A(h,n)). In leukamenin E-treated cells, KFs stained with pK8-Ser73, and pK18-Ser52 antibody decreased significantly (Figure 7A(k,q)), while soluble granules of pK8-Ser73 and pK18-Ser52 increased in HUVECs. These results suggest that leukamenin E-induced phosphorylation at K8-Ser73, and K18-Ser52 was involved in increased keratin solubility and altered polymerization. However, leukamenin E-induced increases in pK8-Ser431 were detected earlier than those of pK8-Ser73 or pK8-Ser52 (Figure 6B). Moreover, pK8-Ser431 response to leukamenin E occurred at lower concentrations than those of pK8-Ser73 or pK8-Ser52 (Figure 6B). These findings indicate that phosphorylation at K8-Ser431 plays a crucial role in leukamenin E-induced increases in soluble fractions of KFs and leads to inhibition of the keratin assembly. Moreover, SPC-induced keratin reorganization is accompanied by keratin phosphorylation, including phosphorylation at K18-Ser 52 and K8-Ser 431. It has been confirmed that K8-Ser431, not K18-Ser52, is the crucial residue for SPC-induced increased soluble fraction of K8/K18 filaments and keratin reorganization by generating mutants of K8-Ser 431 or K18-Ser52 [10]. K18-Ser52 phosphorylation has also been described as a protective signal in response to injury or stress [2,6,47]. It is also assumed that phosphorylation induces release of soluble tetrameric subunits from keratin filaments [48] or that phosphorylation targets depolymerized keratin subunits [49]. The characteristics of K8-Ser431/73 and K18-Ser52 distribution in the cytoplasm suggest that leukamenin E possibly stimulate phosphorylation of soluble keratin subunits and hinder step-by-step assembly of KFs in the cytoplasm. Thus, we propose a working model as shown in Figure 10.

Ser52, Ser73, and Ser431 are major *in vivo* phosphorylation sites in human K8 and K18 [14,50]. The studies have shown that individual phosphorylation sites can be phosphorylated by more than one kinase. Members of the MAPK family have been shown to mediate phosphorylation of keratins. ERK and JNK are involved in pK8-Ser431, and p38 is responsible for pK8-Ser73 [6]. Leukotriene B4 stimulates K8-Ser431 phosphorylation by activating ERK, but SPC stimulates K8-Ser431 phosphorylation by activating JNK [16]. Our findings showed that leukamenin E-stimulated phosphorylation at K8-Ser431, K8-Ser73, and K18-Ser52 was mediated by ERK signaling pathways in PLC and HUVECs.

Busch (2012) proposed that ERK-mediated K8-Ser431 phosphorylation functionally mimics K8 depletion and abolishes the restrictive function of the keratin cytoskeleton on tumor cell migration [10]. Several inducers, including SPC, TPA, and LTB4, induce reorganization of keratin by K8 phosphorylation at Ser431 to promote migration of PANC-1 cells [10,11,16], similar to the effects of epidermal growth factor (EGF). EGF is involved in K8 phosphorylation at Ser431 and promotes cell migration via mitogen-activated protein [37]. This enhanced migration often corresponds to increased activity of the ERK signaling cascade [51]. Nevertheless, the results of other studies contradict the above findings. For example, Tiwari (2017) proposed that the majority of K8 functions were governed by phosphorylation at Ser73 and Ser431 residues based on quantitative phosphoproteomics analysis because they involved system-wide signaling pathways, including cell migration, invasion, and proliferation-associated signaling pathways. Further phenotypic analysis indicated that K8 phosphorylation mutants at Ser73 and Ser431 showed significant inhibition of cell migration and invasion compared to K8-WT [50]. In addition, increased motility was observed in wound-healing scratch assays of human OSCC-derived AW13516 cells, which produced either K8-Ser73A or K8-Ser431A phosphorylation-deficient keratins [19]. Panc-1 cells mainly express K8 and K18 but lack vimentin [18]. Therefore, to exclude the unique effects of drugs on PANC-1 cell migration, we investigated the effects of leukamenin E on the migration of four cell lines, Panc-1, HUVECs, HepG2, and H1299. However, our data clearly show that leukamenin E induced inhibition of migration in all four cell lines (Figure 9, HepG2 and H1299 not shown). Therefore, the correlation between phosphorylation at K8-Ser73 or K8-Ser431 and cell migration needs further investigation.

Some experimental observations provided evidence that keratin phosphorylation is tightly linked to the formation of MDB that are keratin-rich cytoplasmic granular inclusions in hepatocytes. MDBs comprise misfolded hyperphosphorylated K8/K18 with pK8-Ser73/431 and pK18-Ser33 [12]. Phosphorylation at K8-Ser73 is considered a prerequisite for MDB formation [6]. Similarly, K8-Ser73 and K8-Ser431 were found to be hyperphosphorylated in acetaminophen-induced acute liver failure, and this was accompanied by enhanced keratin solubility and perturbed keratin network formation [52]. Thus, leukamenin E-induced keratin phosphorylation at K8-Ser431, K8-Ser73, and K18-Ser52 can provide potential models for the study of these diseases. In addition, our findings also suggest that *ent*-kaurane diterpenoid has a risk of liver injury.

## 4. Materials and Methods

### 4.1. Cell Culture

Primary liver carcinoma cells (PLC), human umbilical vein endothelial cells (HUVECs), and human pancreatic cancer cells (Panc-1) were purchased from the Shanghai Cellular Institute of China Scientific Academy (Shanghai, China). The cells were cultured in 1640 supplemented with 10% fetal bovine serum (*v/v*, Si-ji-qing Biotechnology, Hangzhou, China) at 37 °C in a 5% CO_2_ atmosphere with 95% humidity.

### 4.2. Cell Viability and Apoptosis Assay

Leukamenin E crystal was dissolved in DMSO and diluted to the desired concentrations with medium. The concentration of DMSO in the experimental system was < 0.5 μM, which is below the level at which cell growth and cytoskeleton are not affected.

Cell viability was measured by MTT assay according to the manufacturer’s instructions. Acridine orange/ethidium bromide (AO/EB) staining was conducted as previously described to detect apoptosis [53]. Cells (1 × 10^5^ cells/mL) in the exponential growth phase were seeded on coverslips in 6-well plates for 24 h, after which they were treated with different concentrations of leukamenin E for 24 h. The stained cells were immediately examined using a fluorescence microscope (Leica DMI4000B), and at least 300 cells of each cell type were examined. The following formula was used to count the apoptosis rates:Apoptosis rate = (EA+LA)/(V+N+ EA+LA)×100%(1)

EA: early apoptosis cells; LA: late apoptosis cells; V: viable cell; N: necrotic cells.

### 4.3. Cell Spreading and Well-Spread Treatment

The experimental method reported by literature was used in this study [26]. Briefly, spreading or well-spread cells were treated with different drugs (leukamenin E, cytochalasin B, or colchicine). In the spreading cells experiment, PLC and HUVEC (1 × 10^5^ cells/mL) in the exponential growth phase were trypsinized and re-suspended in serum-free RPMI (Roswell Park Memorial Institute)-1640 for 30 min. The cells were then seeded in serum containing 2.0 μM leukamenin E, 4.2 μM cytochalasin B or 0.1 μM colchicine. At 4 h, 8 h, and 12 h after seeding, the average area of cell spreading was assessed by measuring 30 cells using an area measurement tool (Image J), and cytoplasmic filaments were observed by immunofluorescence assay or K8/K18 phosphorylation was detected by Western blotting assay. In the well-spread cells experiment, after seeding for 24 h, well-spread cells were treated with 2.0 μM leukamenin E for 24 h, 8.4 μM cytochalasin B alone for 24 h, 0.33 μM colchicine alone for 2.5 h, or a combination of all three drugs, after which cytoplasmic filaments were observed by immunofluorescence assay or K8/K18 phosphorylation was detected by Western blotting.

### 4.4. Immunofluorescence Assay

#### 4.4.1. KFs Staining

After the indicated stimulation, cells were fixed with cold methyl alcohol for 5 min at −20 °C. Cells were then permeabilized in phosphate buffer solution (PBS, pH7.4) containing 5% goat serum and 0.1% Triton X-100 for 60 min at room temperature. Next, the monoclonal antibodies anti-keratin-18, anti-keratin-8, anti-keratin-8-Ser431, anti-keratin-8-Ser73, anti-keratin-18-Ser52 (Abcam, Cambridge, UK), or anti-pan-keratin (Cell Signaling Technology, USA) were added overnight at 4 °C. Cells were then incubated with FITC (fluorescein isothiocyanate isomer)-conjugated or Alexa Flour 594-conjugated secondary antibody (Santa Cruz Biotechnology, Dallas, TX, USA) for 1 h without light. After washing with PBS, imaging was performed using a fluorescence microscope (Leica Microsystems, Wetzlar, Germany) equipped with a 100× objective.

#### 4.4.2. MTs Staining

After the indicated stimulation, cells were fixed with 4% formaldehyde for 15 min at room temperature, then permeabilized in PBS (pH7.4) containing 0.1% Triton X-100 for 15 min at room temperature. Next, monoclonal α-tubulin antibodies (Cell Signaling Technology, USA) were added for 1 h at 37 °C. Cells were then incubated with FITC-conjugated secondary antibody (Santa Cruz Biotechnology, Dallas, TX, USA) for 1 h without light. After washing with PBS, cells were imaged as described above.

#### 4.4.3. MFs Staining

After the indicated stimulation, cells were fixed with 4% formaldehyde for 15 min at room temperature and then permeabilized in 0.15% Triton X-100 in PBS for 30 min. Next, cells were incubated in rhodamine-conjugated phalloidin (Sigma-Aldrich, China) for 1 h at 37 °C. After washing with PBS, cells were imaged as above.

### 4.5. Western Blot

Cellular extracts were prepared in lysis buffer (pH7.4, Bestbio, Shanghai, China) for Western blotting and centrifuged at 18,000 g for 15 min at 4 °C. Supernatants were then collected, and protein concentrations were determined using a BCA protein kit (Takara, Tokyo, Japan). Next, equal amounts of protein were loaded and separated on SDS-PAGE gels and then transferred to PVDF membranes (Millipore, USA). After blocking in 5% non-fat milk for 2 h, membranes were incubated with specific primary antibodies overnight at 4 °C. Subsequently, membranes were incubated with secondary antibodies and then incubated with horseradish peroxidase-labeled secondary antibody. The immunoreactive bands were visualized using a ChemiDoc XRS System (BioRad Laboratories, USA). The following monoclonal antibodies were used for Western blotting: anti-pan-keratin (Cell Signaling Technology, USA), anti-keratin-18 (Abcam, Cambridge, UK), anti-keratin-8-Ser431 (Abcam, Cambridge, UK), anti-keratin-8-Ser73 (Abcam, Cambridge, UK), anti-keratin-18-Ser52 (Abcam, Cambridge, UK), anti-α-tubulin (Cell Signaling Technology, USA), anti-p38 (Cell Signaling Technology, USA), and anti-phosphor-p38 (Cell Signaling Technology, USA). Polyclonal anti-keratin-8 antibodies were from CusAb (China), while polyclonal anti-ERK antibody and anti-phosphor-ERK antibody were from Cell Signaling Technology (USA).

To determine keratin solubility, Keratin cytoskeletal preparations were performed with slight modifications according to the method of Beil M. et al. [18]. Briefly, cells treated with or without leukamenin E were lysed in a lysis buffer containing 1% NP40, 5 mM EDTA, 0.5 µg ml^−1^ okadaic acid, 10 µM aprotinin, and 10 µM leupeptin for 15 min on ice. Lysates were subjected to centrifugation at 18,000× *g* for 15 min at 4 °C. The supernatant and the pellet were then collected as two test fractions. Subsequently, the pellet was suspended in the same solubilization buffer and gently homogenized by ultrasonic. Finally, both fractions were analyzed by SDS–PAGE and then transferred to PVDF membranes. The membranes were successively incubated with specific primary antibodies and secondary antibodies. 

### 4.6. Transwell Migration and Wound Healing Assay

Transwell migration and wound healing assays were performed according to the method described by literature [54], with slight modifications. Briefly, 100 μL serum-free 1640 media containing 1 × 10^5^ cells that had been treated with leukamenin E for 1.0 h were seeded onto the inserts with a pore size of 8.0 μm. The bottom inserts were then filled with 500 μL of complete cell culture media as a chemoattractant. After 24 h, the filter membranes of inserts were fixed and stained with crystal violet, and nonmigratory cells were removed by cotton swabs. The underside images of inserts were captured using an Olympus inverted microscope (20×). Cell migration was quantified by counting the number of cells in five random fields, and the inhibitory percentage was performed in relation to the control cells.

Wound healing assays were performed according to the method described by literature [54], with slight modifications. Briefly, monolayer confluent cells plated in 24-well plates were scratched with a 10 μL plastic pipette tip across the cell surface. Floating cells were then washed away using warm PBS. Next, mitomycin-C (final concentration: 10 mg/mL) was added to the plates to inhibit cell proliferation, after which they were rinsed with medium. The wound healing process was monitored with an inverted light microscope (Olympus, Japan), and the cellular migratory capacity was defined as the rate of wound closure (%), which was analyzed using the ImageJ 1.52a software [54].

### 4.7. Materials

Leukamenin E was obtained from *Isodon racemosa* (Hemsl) Hara in our lab [55]. ERK inhibitor (U0126), crystal violet, and acridine orange/ethidium bromide (AO/EB) were obtained from Sigma. Mitomycin C was supplied by AbMole.

### 4.8. Statistical Analysis

The cell spreading area was determined by tracing the cell edge using the area measurement tool in the ImageJ software (Bethesda, MD, USA). The intensities of the Western blot bands were quantified using the ImageJ software (Bethesda, MD, USA). All experiments were repeated three times. All *p*-values were obtained using one-way analysis of variance (ANOVA), and all graphs were generated in Origin8.5. In all figures, * indicates *p* < 0.05 and ** indicates *p* < 0.01.

## 5. Conclusions

Phosphorylation of keratins is involved in the precise dynamic regulation of the assembly–disassembly of KFs. Keratin phosphorylation contributes to regulating the switch between the filamentous and soluble keratin pool [6,56]. The understanding of KF regulation and function relies on multi-faceted approaches that include reagent development to track and modulate unique phospho-epitopes, and other models [6,14,40]. Small molecular compounds that stimulate site-specific phosphorylation of keratin can be used as molecular probes to study regulation and function of KFs. Our experimental data confirm that leukamenin E-induced phosphorylation at K8-Ser73/431 and K18-Ser52 was involved in increased keratin solubility and blocks the assembly of keratin filaments networks through ERK activation. Leukamenin E may interfere with initiation of KFs assembly at the periphery of the cell and block the step-by-step assembly of KFs and the formation of a new KFs network and then inhibit turnover of KFs in HUVECs. This study is the first to report the novel pharmacological target and the effects of leukamenin E, an *ent*-kaurane diterpenoid. Based on the very rich natural resources of *ent*-kaurane diterpenoids, they will be important to drug discovery, as well as to molecular probes used to investigate KF regulation and function.

## Figures and Tables

**Figure 1 ijms-21-03164-f001:**
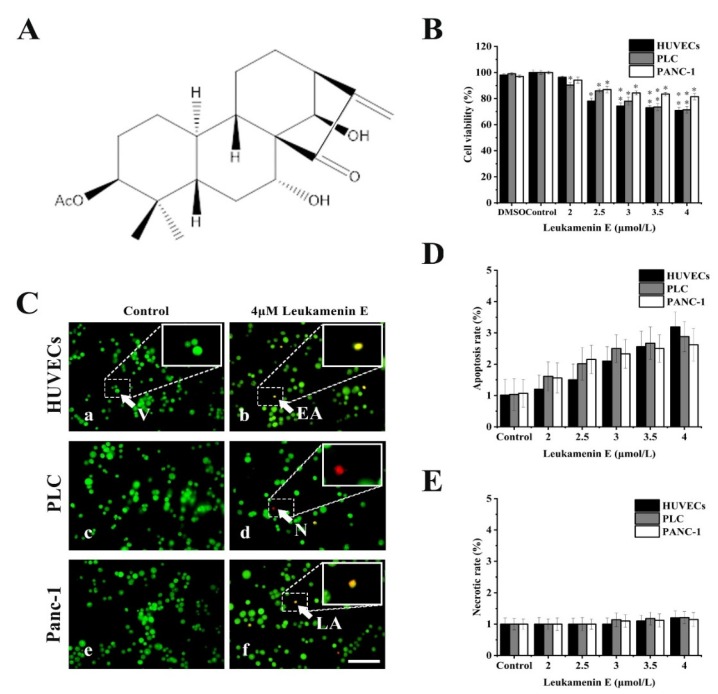
Effects of leukamenin E on proliferation and apoptosis in human umbilical vein endothelial cells (HUVECs), primary liver carcinoma cells (PLC), and Panc-1 cells. (**A**) Chemical structure of leukamenin E. (**B**) Cell viability assay of HUVECs, PLC, and Panc-1 treated with leukamenin E (2 μM-4 μM) for 24 h based on MTT. (**C**) Evaluation of apoptosis of HUVECs (a,b), PLC (c,d), and Panc-1 (e,f) cells treated with leukamenin E by acridine orange/ethidium bromide (AO/EB) double staining. Four types of cells were observed by AO/EB staining: viable cells (V), early apoptotic cells (EA), necrotic cells (N), and late apoptotic cells (LA), as indicated by the arrows in the pictures. Bar, 40 µm. (**D**) Apoptotic rate in HUVECs, PLC, and Panc-1 cells treated with leukamenin E for 24 h. (**E**) Necrotic rate in HUVECs, PLC, and Panc-1 cells treated with leukamenin E for 24 h. Images represent typical cells from at least three independent experiments. The data shown represent the means ± standard deviation (S.D.) of three independent experiments. * *p* < 0.05, νersus the control group.

**Figure 2 ijms-21-03164-f002:**
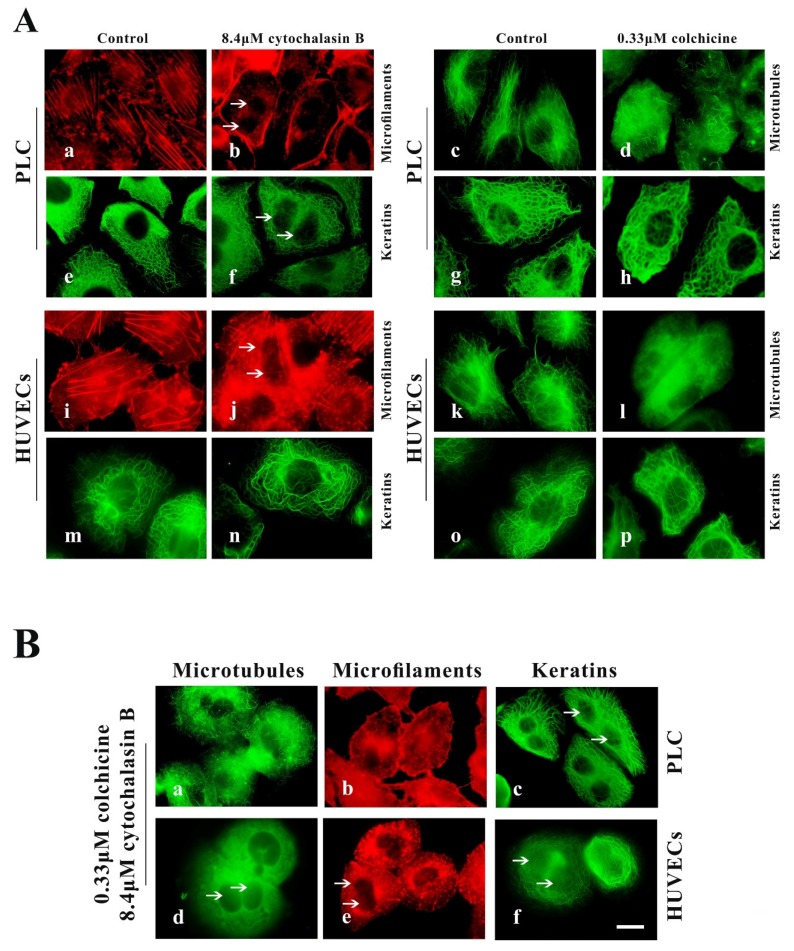
Cytochalasin B and colchicine cause the depolymerization of microfilament (MFs) and microtubules (MTs), respectively, in well-spread PLC and HUVECs but do not distinctly alter the keratin filament (KF) network. (**A**) Representative images of cell morphology and three cytoskeleton components in well-spread PLC and HUVECs treated with cytochalasin B (b,j,f,n) or colchicine (d,l,h,p) or DMSO (control, a,i,e,m,c,g,k,j,o). 1000×, Bar, 10 µm. (**B**) Representative images of cell morphology and three cytoskeleton components in well-spread PLC (a,b,c) and HUVECs (d,e,f) treated with cytochalasin B and colchicine. 1000×, Bar, 10 µm. The binucleate cells caused by cytochalasin B are indicated by arrows. Images represent typical cells from at least three independent experiments.

**Figure 3 ijms-21-03164-f003:**
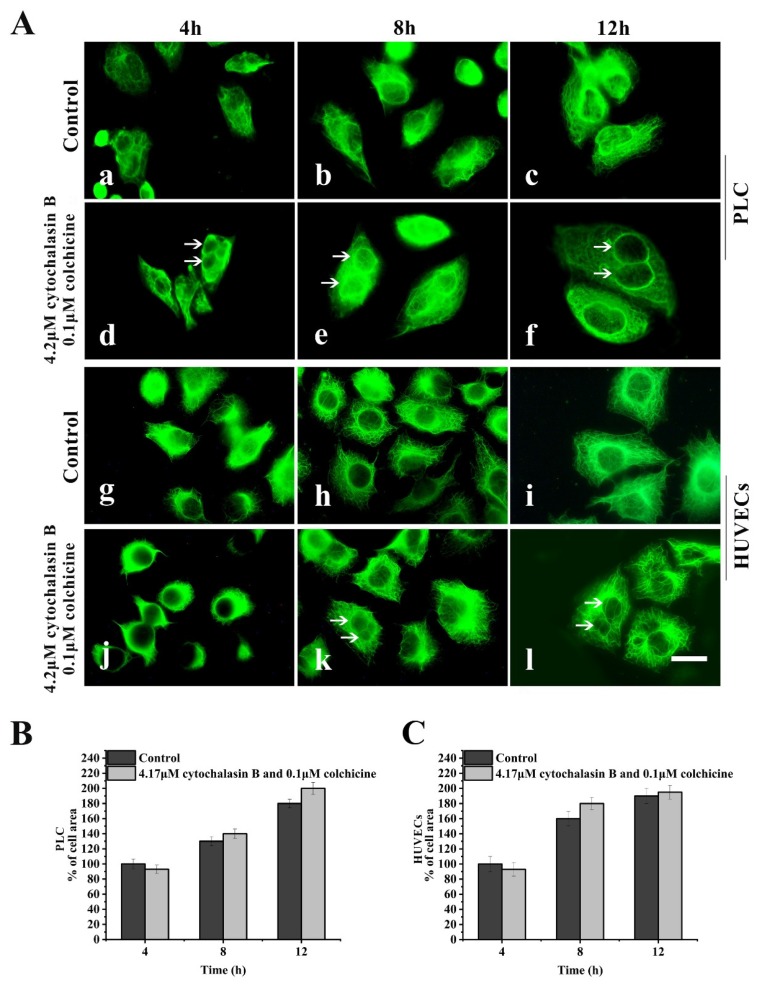
Effects of cytochalasin B and colchicine on KFs reassembly and cell spreading. (**A**) Representative images of KF network reassembly and cell morphology at 4–12 h after suspended PLC and HUVECs were seeded in media with cytochalasin B and colchicine (d,e,f,j,k,l) or DMSO (control, a,b,c,g,h,i), showing that treatment with cytochalasin B and colchicine does not inhibit KFs reassembly and cell spreading. 1000×, Bar, 10 µm. (**B**,**C**) Areas of cell spreading were measured at 4 h, 8 h, and 12 h after suspended HUVECs or PLC were seeded in medium-containing cytochalasin B and colchicine or in control medium (DMSO). The binucleate cells caused by cytochalasin B are indicated by arrows. Images represent typical cells from at least three independent experiments. The data shown represent the means ± standard deviation (S.D.) of three independent experiments.

**Figure 4 ijms-21-03164-f004:**
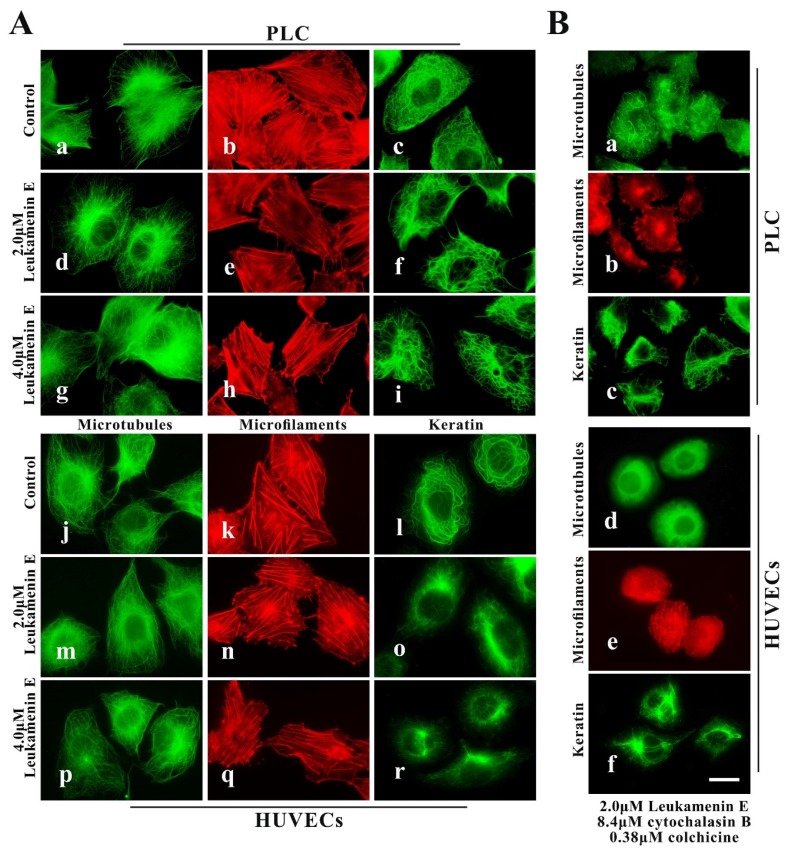
Leukamenin E destroys the KF network in HUVECs and PLC. (**A**) Representative images of cell morphology and three cytoskeleton components in well-spread HUVECs and PLC treated with 2.0 μM leukamenin E (d,m,e,n,f,o) or 4.0 μM leukamenin E (g,p,h,q,i,r) or DMSO (control, a,j,b,k,c,l) for 24 h, showing that leukamenin E destroys the KF network (f,I,o,r) but slightly alters MFs (e,h,n,q) and MTs (d,g,m,p). 1000×, Bar, 10 µm. (**B**) Representative images of cell morphology and three cytoskeleton components in well-spread HUVECs and PLC treated with a combination of leukamenin E, cytochalasin B, and colchicine for 24 h, showing that leukamenin E destroys the KF network (c,f) independently of MFs (b,e) and MTs (a,d). 1000×, Bar, 10 µm. Images represent typical cells from at least three independent experiments.

**Figure 5 ijms-21-03164-f005:**
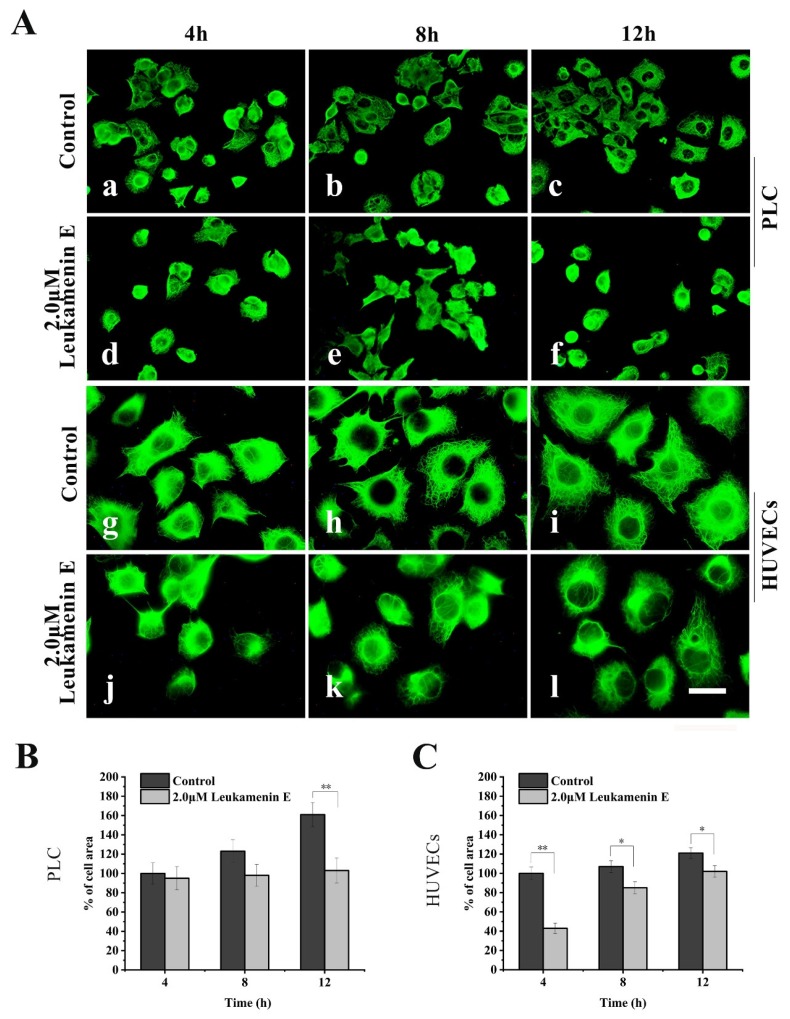
Leukamenin E inhibits KFs reassembly and cell spreading in HUVECs and PLC. (**A**) Representative images of KF network reassembly and cell morphology at 4–12 h after the suspended PLC and HUVECs were seeded in media with leukamenin E (d,e,f,j,k,l) or DMSO (control, a,b,c,g,h,i), showing the inhibition effect of leukamenin E on KFs reassembly and cell spreading. 1000 ×, Bar, 10 µm. (**B**,**C**) Areas of cell spreading were measured at 4 h, 8 h, and 12 h after suspended HUVECs or PLC were seeded in medium-containing leukamenin E or DMSO (control). Images represent typical cells from at least three independent experiments. The data shown represent the means ± standard deviation (S.D.) of three independent experiments. ** p* < 0.05, ** *p* < 0.01, νersus the control group.

**Figure 6 ijms-21-03164-f006:**
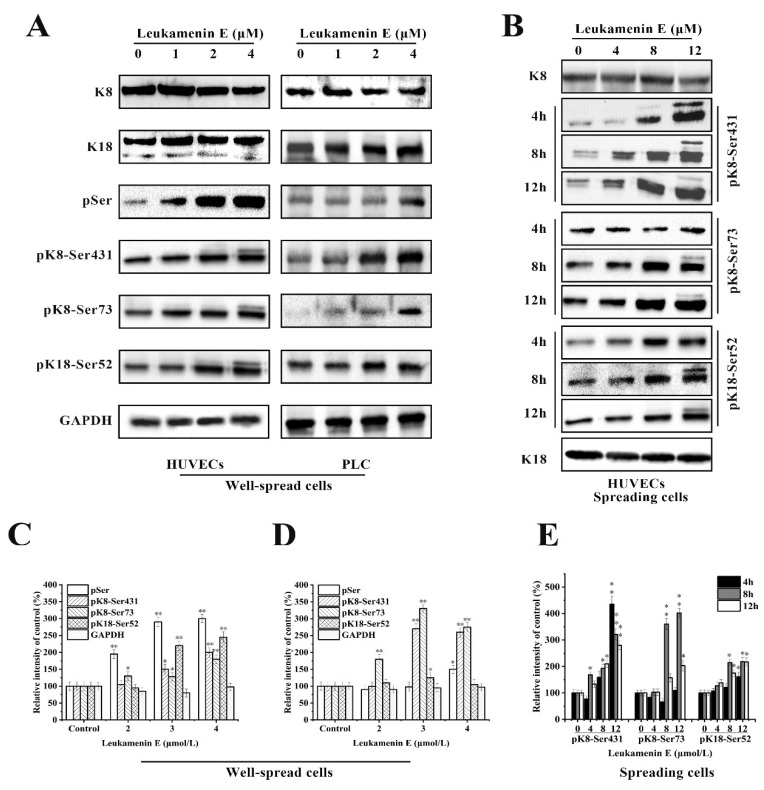
Effects of leukamenin E on keratin phosphorylation in PLC and HUVECs. (**A**) Leukamenin E stimulates phosphorylation of K8 at Ser73 and Ser431 and K18 at Ser52 but does not alter the expression of K8 and K18 in well-spread cells. (**B**) Leukamenin E stimulates phosphorylation of K8 at Ser73 and Ser431 and K18 at Ser52 in spreading cells. (**C**–**E**) Quantification of keratin or phosphorylated keratin by grey level analysis. The data represent the means ± standard deviation (S.D.) of three independent experiments. * *p* < 0.05, ** *p* < 0.01 versus control group.

**Figure 7 ijms-21-03164-f007:**
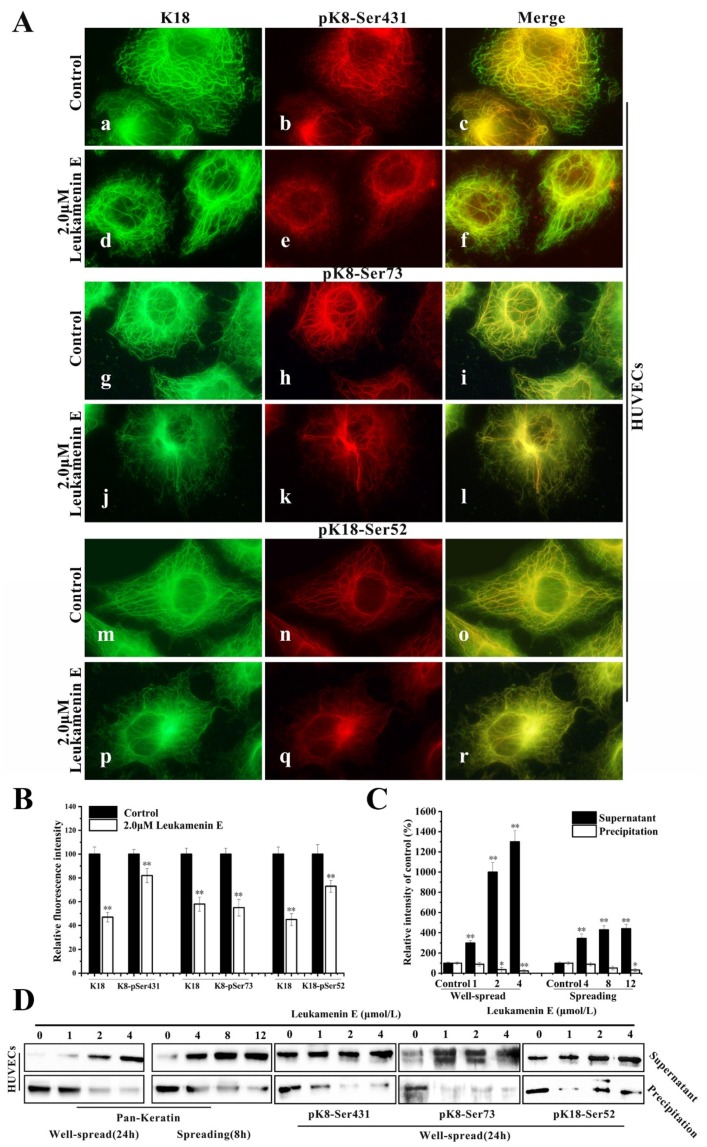
Leukamenin E induces phosphorylation at K8-Ser431, K8-Ser73 and K18-Ser52 to increase keratin solubility in HUVECs. (**A**) Immunofluorescence imaging shows that leukamenin E leads to increased phosphorylated KFs fraction from keratin network. Keratin immunostaining was performed using the pK8-Ser431 antibody (b,e), pK8-Ser73 (h,k) antibody, pK18-Ser52 antibody (n,q), and K18 antibody (a,d,g,j,m,p). The mergers are shown in c, f, i, l, o and r, respectively. 1000×. (**B**) Graph depicts the quantification of K8 or phosphorylated K8 at Ser431/Ser73/Ser52 by fluorescence intensity analysis. (**C**) Protein content in supernatant and precipitation was determined by grey level analysis. (**D**) Cells were subsequently lysed, and the keratin content in the supernatant and pellet was analyzed by Western blotting using pan-keratin /pK8-Ser431/pK8-Ser73/pK18-Ser52antibody, which showed that leukamenin E increases soluble fraction of KFs or phosphorylated keratin in well-spread cells. Images represent typical cells from at least three independent experiments. The data shown represent the means ± standard deviation (S.D.) of three independent experiments. * *p* < 0.05, ** *p* < 0.01, νersus the control group.

**Figure 8 ijms-21-03164-f008:**
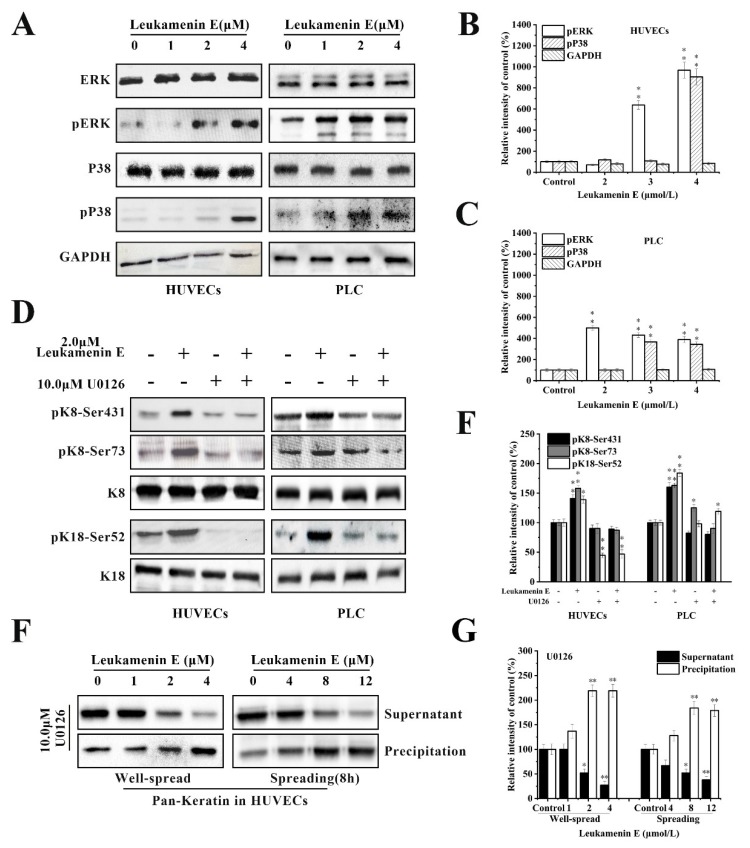
Extracellular signal-regulated kinases (ERK) mediates leukamenin E-induced K8/18 phosphorylation and increased keratin solubility in HUVECs and PLC. (**A**–**C**) Leukamenin E-induced activation of ERK and P38 phosphorylation in HUVECs and PLC and their quantitative analysis. (**D**) ERK mediates leukamenin E-induced K8/18 phosphorylation. HUVECs and PLC were treated with 10 μM U0126 for 1 h followed by incubation with 2.0 μM leukamenin E for 24 h. Western blotting was performed using pK8-Ser431, pK8-Ser73, and pK18-Ser52 antibody. (**E**) Graph depicts the quantification of keratin by grey level analysis. (**F**) ERK mediates leukamenin E-induced increase of keratin solubility. Well-spread HUVECs were treated with 10 μM U0126 for 1 h followed by incubation with 2.0 μM leukamenin E for 24 h or the spreading HUVECs were treated with 10 μM U0126 for 1 h followed by incubation with leukamenin E at the indicated concentrations for 8 h. WB was performed using Pan-K antibody. (**G**) Graph depicts the quantification of keratin by grey level analysis. The data shown represent the means ± standard deviation (S.D.) of three independent experiments. * *p* < 0.05, ** *p* <0.01, νersus the control group.

**Figure 9 ijms-21-03164-f009:**
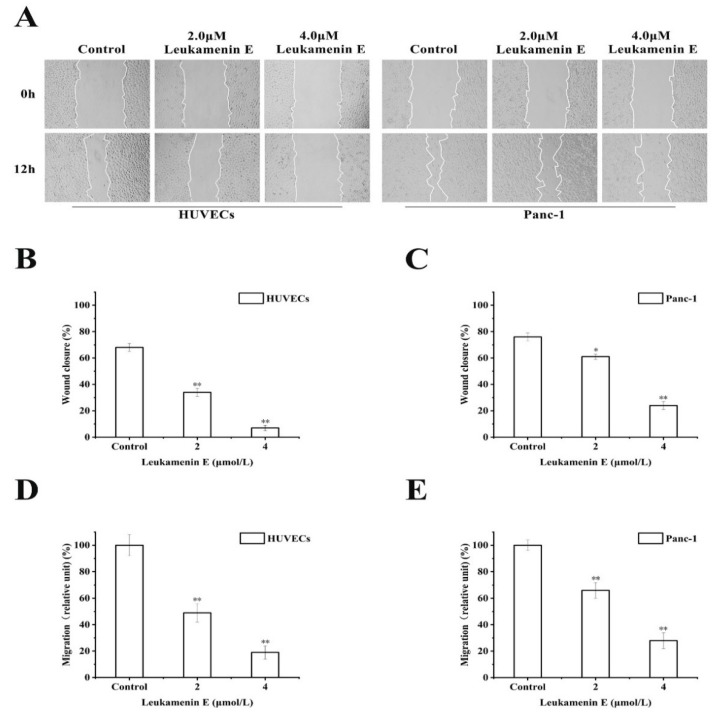
Leukamenin E inhibits migration of HUVECs and Panc-1 cells. (**A**) Scratch wound healing assays were performed to detect the migration of the indicated cells. Images of the scratch wound were captured after 12 h of culture with or without leukamenin E. 100× multiplication. (**B**,**C**) Graph quantifying the rate of wound closure in HUVECs and Panc-1 cells. (**D**,**E**) HUVECs and Panc-1 cells were incubated with 2.0 μM and 4.0 μM. The number of cells migrating through size-limited pores was determined by inverted microscopy. The results are presented as means ± SEM. The data shown represent the means ± standard deviation (S.D.) of three independent experiments. ** p* < 0.05, *** p* < 0.01, νersus the control group.

**Figure 10 ijms-21-03164-f010:**
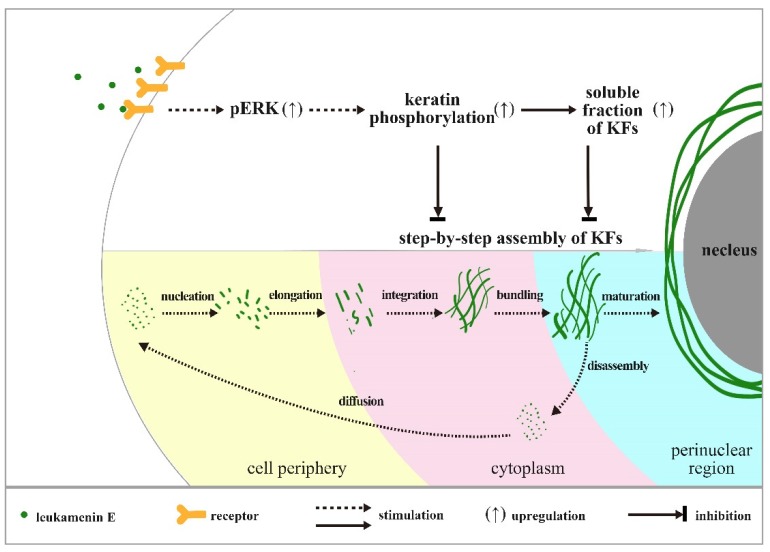
Working Model. Our data are consistent with a model in which leukamenin E-induced phosphorylation at K8-Ser73/431 and K18-Ser52 was involved in increased soluble fraction of KFs and altered polymerization. Leukamenin E may interfere with initiation of KFs assembly at the periphery of the cell and block step-by-step assembly of KFs and the formation of a new KFs network and then inhibit turnover of KFs in HUVECs.

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
