# Peer review of "Leukamenin E Induces K8/18 Phosphorylation and Blocks the Assembly of Keratin Filament Networks Through ERK Activation"

_ijms, 2020, doi:10.3390/ijms21093164_

Round 1
Reviewer 1 Report
I acknowledge the efforts of the authors and don't want to delay the publication of the data.
Author Response
Thank you very much for your recognition of our work.
Reviewer 2 Report
The authors have not checked the manuscript before submitting it, so the submitted version has the ”track changes” option on, making a very complex paper as this one is, even more complex. In addition, there are mistakes in the English language as well as inconclusive sentences, which make me feel that the authors just forgot to take them out. This makes the paper very hard to read and understand, very messy altogether.
Regarding the content of the manuscript: one thing that is not fully clear is the aim of this study. Was it to find keratin polymerization inhibitors, to look for anti-cancer drugs or both or other?
In the last two decades there were many papers about the interconnection of the IF network with MF and MT systems, so is there a theory what mechanism allows in this case the keratin network to be independent from the other two? Keratin subunit transport has been demonstrated closely dependent from the MF and MT networks, so to say that depolymerization of MFs and MTs has no effect on the keratin network is a little bit far fetching to me.
As it stands, the paper feels like a lot of work has been done without a clear aim (or has just been badly presented). The Conclusion section is thus quite inconclusive. Instead of providing a wider picture, the authors use it to quote once again their experiments.
All in all, to me this is a paper that needs major revision in style and aims before it can be resubmitted for another round of review.
Author Response
Dear Reviewers,
I sincerely appreciate your professional comments on our manuscript entitled “Leukamenin E induces K8/18 phosphorylation and blocks the assembly of keratin fiber networks through ERK activation” (ijms-774051). The comments were valuable and helped us to revise and improve our paper. We have reviewed the comments carefully and have revised the manuscript accordingly.
The revised portions are marked in blue text, and some additional errors not mentioned by the reviewers have also been rectified. We are prepared to make further revisions if necessary. Point-by-point responses to the reviewers’ comments are provided below.
We look forward to your response.
Sincerely,
Responses to the Reviewers’ Comments
Reviewer’s comment: (The authors have not checked the manuscript before submitting it, so the submitted version has the ”track changes” option on, making a very complex paper as this one is, even more complex. In addition, there are mistakes in the English language as well as inconclusive sentences, which make me feel that the authors just forgot to take them out. This makes the paper very hard to read and understand, very messy altogether.)
Response: Thank you for this suggestion. We have made the suggested corrections.
LetPub (www.letpub.com) has provided the linguistic assistance during the preparation of this manuscript.
Reviewer’s comment: (Regarding the content of the manuscript: one thing that is not fully clear is the aim of this study. Was it to find keratin polymerization inhibitors, to look for anti-cancer drugs or both or other?)
Response: The purpose of our experiments was: (1) to investigate possible modes of action of leukamenin E on keratin filaments, and (2) to find keratin polymerization inhibitors.
Reviewer’s comment: (In the last two decades there were many papers about the interconnection of the IF network with MF and MT systems, so is there a theory what mechanism allows in this case the keratin network to be independent from the other two? Keratin subunit transport has been demonstrated closely dependent from the MF and MT networks, so to say that depolymerization of MFs and MTs has no effect on the keratin network is a little bit far fetching to me.)
Response: There is growing evidence that the three cytoskeletal subsystems interact through direct physical contact mediated by cross-linkers. Keratin organization seems to depend mostly on interactions with MFs (Windoffer R, 2011). Keratin-MT crosstalk also occurs by an unresolved mechanism that involves plectin (Valencia RG, 2013). In many cells types, deconstruction of some types of cytoskeleton filaments is usually accompanied by the collapse of another cytoskeleton. Nevertheless, the studies have shown that the keratin network is a relatively independent cytoskeletal subsystem in epithelial cells. For example, in vivo keratins form IF systems distinct from vimentin IFs and MT systems. This becomes particularly evident after drug-induced destruction of microtubules, which causes vimentin IFs to reorganize while leaving the CK IF system untouched (Figure 1). Our experiments showed that depolymerization of MFs or MTs by the MF-destroying drug cytochalasin B or the MT-destroying drug colchicine, could not concomitantly disrupt KFs in epithelial cells (HepG2, NCI-H1299, Hela, PLC, A549 and HUVECs cells), and the integrity of the KFs network was maintained. However, at higher concentrations the combined treatment of cytochalasin B and colchicine resulted in significant rearrangement of the keratin cytoskeleton, being consistent with literatures (e.g. Loren W., 1983). Some types of epithelial cells are optimal models for investigating the regulation and assembly-disassembly mechanisms of KFs, as well as screening for keratin inhibitors.
(Pictures are available in the pdf attachment)
Figure 1.(Herrmann, 2000)
Windoffer R, Beil M, Magin TM, Leube RE: Cytoskeleton in motion: the dynamics of keratin intermediate filaments in epithelia. J Cell Biol, 2011, 194:669-678
Valencia RG, Walko G, Janda L, Novacek J, Mihailovska E, Reipert S, Andra¨ -Marobela K, Wiche G: Intermediate filamentassociated cytolinker plectin 1c destabilizes microtubules in keratinocytes. Mol Biol Cell, 2013, 24:768-784.
Herrmann, H. and Aebi, U., Intermediate filaments and their associates: multi-talented structural elements specifying cytoarchitecture and cytodynamics. Current Opinionin Cell Biology, 2000, 12: 79-90.
Loren W., Knapp, W., Michael O', Roger H. S, Rearrangement of the Keratin Cytoskeleton after Combined Treatment with Microtubule and Microfilament Inhibitors. J Cell Biol, 1983, 97:1788-1794.
Reviewer’s comment: (As it stands, the paper feels like a lot of work has been done without a clear aim (or has just been badly presented). The Conclusion section is thus quite inconclusive. Instead of providing a wider picture, the authors use it to quote once again their experiments.)
Response: Thank you for this suggestion. we revised the conclusion section of the manuscript.

Round 2
Reviewer 2 Report
Figure 10 showing/describing the working model is unclear with the current letter type and size. The figure legend could also benefit from a more detailed description of what it is showing, especially if the figure mentions signalling pathways, which are not included in the legend.
Author Response
Dear Reviewers,
I sincerely appreciate your professional comments on our manuscript, “Leukamenin E induces K8/18 phosphorylation and blocks the assembly of keratin fiber networks through ERK activation (ijms-774051)”. The comments are very helpful to the revision and improvement of our paper. We have revised the manuscript accordingly. The revised portions are marked in blue text, and some additional errors not mentioned by reviewers have also been rectified. We are prepared to make further revisions if necessary. The responses to the reviewers’ comments are provided below.
We look forward to your response.
Sincerely,
Responses to the Reviewers’ Comments
Reviewer’s comment: (Figure 10 showing/describing the working model is unclear with the current letter type and size. The figure legend could also benefit from a more detailed description of what it is showing, especially if the figure mentions signalling pathways, which are not included in the legend.)
Response: Thank you for this suggestion. In order to improve the resolution of figure, we have redrawn Figure 10 (Working Model) with Photoshop (PS) software. Accordingly, we have added more details of the stepwise assembly of KFs and signaling pathways in Figure 10. The figure legend was added in Figure 10.
This manuscript is a resubmission of an earlier submission. The following is a list of the peer review reports and author responses from that submission.
Round 1
Reviewer 1 Report
The manuscript by Xia et al. investigates the mechanism of action of Leukamenin E on keratin cytoskeleton. The authors demonstrated that Laukamenin E acts specifically on keratin filaments without affecting tubulin and actin cytoskeleton. Moreover the authors demonstrated that Leukamenin E impacts on keratin filament phosphorylation, assembly and organization affecting spreading and migration of cells. The results are interesting and deserve publication on International Journal of Molecular Sciences after answering to the following criticism:
Major points
-The cells shown in Fig.1C are too little to fully appreciate the differences among the different situations and, in particular, the difference between early and late apoptotic cells is not evident. A higher magnification, even only of details, should be used.
-The graphs in Fig.1D and 1E have a X axis scale of 50 while the values reached are less than 5. This scale does not allow to appreciate the possible differences among the different samples, thus I suggest the X axis scale to be at maximum 5
-The authors measure spreading in Fig. 3. It is opinion of this reviewer that spreading should be measured at earlier time points (30 min, 1 h, 2h) after plating in order to better visualize and quantify possible differences.
-Keratin filaments alterations are evident in HUVEC's (Fig.4Ao,r) but not in PLC cells (Fig.4Af,i). The authors should at least comment on this difference.
-It will be important to have a figure with a scheme depicting the different mechanism of action of Leukamenin E.
-Methods should be described in more details.
Minor points
-PLC cells in Fig.5 are taken at a too low magnification. To better appreciate differences in spreading they should be shown at a higher magnification.
Author Response
Dear Reviewers,
I sincerely appreciate your professional comments on our manuscript, “Leukamenin E induces K8/18 phosphorylation and blocks the assembly of keratin fiber networks through ERK activation (ijms-716872).” We found these comments very helpful to the revision and improvement of our paper. We have reviewed each one carefully and have revised the manuscript accordingly.
The revised portions are marked in red, and we took this opportunity to make additional improvements not mentioned by the reviewers. We are prepared to make further revisions if necessary. Point-by-point responses to the reviewers’ comments are provided below.
We look forward to your response.
Sincerely,
Responses to the Reviewers’ Comments
Reviewer’s comment: (The cells shown in Fig.1C are too little to fully appreciate the differences among the different situations and, in particular, the difference between early and late apoptotic cells is not evident. A higher magnification, even only of details, should be used.)
Response: It is essential to show as many cells as possible and there was a low rate of apoptosis, as shown in Fig. 1C, so it is difficult to present the details of the stained cells at any lower magnification (10×10). Considering the reviewer’s suggestion, in order to show the difference between early and late apoptotic cells, we added a small window to show different types of cells in the top right corner of the pictures in Fig. 1C.
Reviewer’s comment: (The graphs in Fig.1D and 1E have a X axis scale of 50 while the values reached are less than 5. This scale does not allow to appreciate the possible differences among the different samples, thus I suggest the X axis scale to be at maximum 5)
Response: Thank you for this suggestion. We have made the suggested corrections.
Reviewer’s comment: (The authors measure spreading in Fig. 3. It is opinion of this reviewer that spreading should be measured at earlier time points (30 min, 1 h, 2h) after plating in order to better visualize and quantify possible differences.)
Response: The cells cultured in vitro were allowed to reattach to slide for at least 3 h after plating, so the spreading was measured at 4 h after inoculation. If the spreading was measured any earlier (30 min, 1 h, 2 h), a large number of cells would fall off the slide after immunofluorescence processing.
Reviewer’s comment: (Keratin filaments alterations are evident in HUVEC's (Fig.4Ao,r) but not in PLC cells (Fig.4Af,i). The authors should at least comment on this difference.)
Response: Thank you for this suggestion. Accordingly, we have added a comment on this difference to the revised manuscript in 2.2.2 section.
We have observed that the keratin filaments in control PLC cells are thick and dense, while the keratin filaments in control HUVECs cells are relatively sparse, so we believe that the keratin filament network of HUVECs cells may be more damaged than that of PLC cells when treatment with leukamenin E at the same concentration. It may also be because of differences in their molecular metabolism.
Reviewer’s comment: (It will be important to have a figure with a scheme depicting the different mechanism of action of Leukamenin E.)
Response: Thank you for this suggestion. We have added a diagram depicting the mechanism of action of leukamenin E is to the revised manuscript in the results section.
Reviewer’s comment: (Methods should be described in more details.)
Response: The experimental methods used in this manuscript are the usual methods used in pharmacological research. After careful examination of the description of the experimental method, we found that the text in Section 4.3 (Cell spreading and well-spread treatment) may not have been sufficiently detailed. The experimental method used in this section is described in great detail in our published literature (Zhu J et al., 2019), which is now cited in Section 4.3.
In addition, we supplemented the method for the detection of keratin solubility in Section 4.5.
Reviewer’s comment: (PLC cells in Fig.5 are taken at a too low magnification. To better appreciate differences in spreading they should be shown at a higher magnification.)
Response: The spreading of PLC cells is slower than that of HUVEC cells, so the area of PLC cells is smaller than that of HUVECs cells at the same magnification. Considering the reviewer's suggestion, the PLC cells in Fig. 5 have now been properly magnified using Photoshop software.
Reviewer 2 Report
The authors propose that the natural compound Leukamenin E leads to keratin K8/K18 phosphorylation at K8-Ser73/Ser431 and K18-Ser52 via MAPK pathway. They further claim that Leukamenin E-induced keratin phosphorylation increases the soluble, phosphorylated K8/K18 pools leading to inhibited spreading and migration of the cells. This may facilitate keratin dissociation and block keratin assembly. Recently, Zhu J et al. investigated the effect of Leukamenin E onto KFs using HepG2 cells and NCI-H1299 cells. In the present manuscript the authors use PLC, HUVEC and PANC-1 cells.
Finding a compound that is able to manipulate keratin dis/assembly independently of other major cytoskeletal components (microfilaments, microtubules) would be a powerful tool for understanding KF assembly and organization. Unfortunately, the current ms adds very little to demonstrate that LeuE acts selectively on keratins. There are a number of fundamental issues that require experimental attention:
Given its interference with the activity of major kinases, selectivity for keratins appears highly unlikely. In order to get some solid evidence that the above P-sites are directly responsible for keratin reorganization/solubilization and for the effects studied, the authors need to generate the corresponding P-deficient and P-mimetic mutants and study them in at least one of the cell lines used. This will show to which extent LeuE is selective and to which extent the P-sites identified are responsible for effects in spreading, migration and cell survival. As essential control, knockdown of K8 and K18 has to be performed in at least one cell line. The criterion “solubility” is neither well defined nor investigated. Fractionation in cytoskeletal buffers (1 % TX100, 1 % TX100 plus 1.5 M KCl) would help. It needs to be examined whether LeuE can directly bind to K8 and/or K18 and thereby alter their solubility/biophysical properties. This can be done in vitro using purified keratins. The fixation conditions described appear strange. Following MeOH fixation (which permeabilizes membranes), detergent extraction is performed. This is likely to affect a number of proteins by solubilizing them. I strongly recommend to omit the latter step and to compare MeOH with formalin/detergens fixation protocols.
Overall, most of the results are overinterpreted and unfortunately not convincing in the current manuscript.
Points in detail:
Fig. 1: The corresponding vehicle control is missing. Although the corresponding DMSO concentration is below 0.05 µM one needs to show that there is no DMSO effect (using the highest concentration). Higher concentrations of Leukamenin E lead to the inhibition of proliferation (e.g. 3.5 and 4 µM), which is shown for three cell lines (HUVEC, PLC and PANC-1), Fig. 1B. Apoptosis rates and necrosis seem to be unaffected by Leukamenin E. It is hard to discriminate between early and late apoptotic cells (Fig. 1C) as they all appear yellow. Detailed images would be helpful. In addition, the graphs in Fig. 1D and E need to be changed as the individual bars are not really visible (y-intercept of max. 20%). It appears that 4 µM Leukamenin E increases apoptosis in all three tested cell lines (Fig. 1D). The authors conclude that 2-4 µM Leukamenin E are suitable concentrations for further experiments. However, in Fig. 6B, 7B and 8F even higher concentrations of the compound were used (8 and 12 µM) in HUVEC without testing the toxic effects. Washouts and recovery after 24h are needed.
Fig. 2: It needs to be better explained why the authors distinguish between spread and spreading cells. In this figure it is hard to believe that depolymerization of MTs and MFs has absolutely no impact on keratin filaments, also because the authors show no co-stainings. In Fig. 2A the Cytochalasin B and Colchicine effect is shown by stainings of the individual cells. For the impact onto keratin filaments completely different cells are shown. In Fig. 2B the corresponding control (DMSO) is missing and again co-staining would be helpful to convince the conclusion.
Fig. 3: The authors claim that in the spreading experiment the assembly of MTs and MFs was inhibited in Cytochalasin B- and Colchicine-containing medium without showing any staining for MTs or MFs. Furthermore it appears that 12h Cytochalasin B and Colchicine treatment affects keratin filament organization (Fig. 3A f and l). So it is not convincing to conclude that the KF network in PLC and HUVEC is independent of MFs and MTs.
Fig. 4: Again the corresponding DMSO control for the highest concentration has to be shown (Fig. 4A). In Fig. 4B no loading control is shown. Co-staining of the networks as well as quantification would be helpful (e.g. keratin network normal versus bundled for 4 µM compound compared to DMSO).
Fig. 5: The authors claim that in this cell spreading experiment the cells treated with Leukamenin E showed significant inhibition of keratin assembly. The images for the PLCs need to be shown in higher magnification (details in Fig. 5A would be helpful). There could be the possibility that Leukamenin E indirectly alters the ability to spread. But as one can clearly see keratin filaments, it is hard to believe that keratin assembling is inhibited by the compound. In vitro keratin assembly experiments in the presence of Leukamenin E could show a putative inhibitory effect. In addition, biochemical fractionation into soluble and insoluble keratins is needed.
Fig. 6: In this figure Xia et al. investigate, whether Leukamenin E treatment causes keratin K8/K18 phosphorylation by the use of site-specific phospho-antibodies. Fig. 6A and B together confuse the reader and some of the WBs appear over-exposed or the contrast is too high. Again the authors distinguish between well-spread and spreading cells (why?). However, in the spreading cell experiments just HUVECs are shown (Fig. 6B), PLCs not (why?). There is no information in the text, why the authors also used an unspecific pSer antibody in Fig. 6A. The quantification in Fig. 6C and D would be clearer, if the phospho-keratin signal would be normalized against the corresponding unphosphorylated keratin signal (e.g. pK8-Ser431 against K8 signals). The signals of unphosphorylated K8 and K18 or a loading control are even completely missing in Fig. 6B. Again I am wondering, why the authors used so high concentrations (8-12 µM in Fig. 6B)? Vehicle control (especially for the 12 µM treatment) is missing. Based on the WBs in Fig. 6B the authors claim that K8-Ser431 occurred earlier than that of K8-Ser73 and K18-52 in Leukamenin E-treated cells. However, to show a time- and dose-dependency all samples should be run on one gel. Furthermore, it is not clear why there is less phosphorylated K8-Ser73 at 8h (0 and 4 µM) compared to 4h? And why there is also less phosphorylated K18-Ser52 at 4h 8 µM treatment compared to 0, 4 or 12 µM treatment? Quantification in Fig. 6E should be changed like in Fig. 6C and D.
Fig. 7: The authors performed double-stainings using K18 together with K8-Ser431, -Ser73 or K18-Ser52 antibodies to show changes in the phospho-K8/K18 signals after Leukamenin E treatment. The described changes are hard to see; detailed and confocal images would be helpful. As the bundled keratin filaments appear brighter (Fig. 7A h), it could be that the filaments at the cell periphery appear darker. Furthermore, the control shows also a cell with keratin bundling around the nucleus (Fig. 7A m). In Fig. 7B the vehicle control is missing as well as a loading control or Ponceau S staining. How the two keratin pools were made (supernatant, precipitate); there is no information in the methods sections. Why the authors used a Pan-Keratin antibody in Fig. 7B and no K8 or K18 antibody? This signal could be better compared with the phospho-signals of the WBs from Fig. 7D. The quantification in Fig. 7C does not reflect the WBs in B, e.g. compare supernatant signals of 0 and 1/4 µM of well-spread to spreading cells. The signal for the spreading cells at 4 µM appears higher compared to 1 µM in well-spread cells, but the quantification shows nearly equal relative intensities? Also in Fig. 7D a loading control is missing and some of the blots appear over-exposed.
Fig 8: The authors next determined whether ERK is involved in Leukamenin E-induced K8/K18 phosphorylation as it is already known that this kinase can phosphorylate K8. They show that the compound stimulated ERK phosphorylation in a dose-dependent manner. Here I would also normalize the phospho-signals against unphosphorylated protein signals (Fig. 8B and C). Fig. 8D shows no unphosphorylated K8/K18 or a loading control. It is also necessary to show that U0126 really blocks MAPK pathway (e.g. phosphorylation of downstream p38) under this conditions. Using PLCs: there is no effect onto pK8-Ser73 and pK18-Ser52 upon ERK inhibitor use. Furthermore, there is no text information in the manuscript about Fig. 8E-G?
Fig. 9: The authors next investigated the impact of Leukamenin E onto cell migration. Using a transwell as well as a wound healing assay they claim that the compound induced an inhibitory effect on migration of HUVEC and Panc-1 cells. It was shown by others that small compounds like TPA or SPC can promote migration of Panc-1 cells. This could be a good positive control and added to the experiments. Also due to the fact, that proliferation rates were slightly decreased (Fig. 1B, compare control to 4 µM Leukamenin B treatment of Panc-1 cells) the inhibitory migration effect could be due to proliferation effects. To avoid this, a proliferation inhibitor could be also used in the wound healing assay. Minor point: show the gap closure as long as the control is completely closed (longer time points).
Discussion:
It is not convincingly shown that Leukamenin E exerts its effects on keratin filaments independently of other major cytoskeletal components. The authors propose that Leukamenin E can specifically inhibit the assembly of KFs, however a lot of KFs are seen in all the images after compound treatment. The statement “Leukamenin E presumably hinders KFs assembly in vivo because it can covalently link to synthetic peptides in vitro” needs to be better explained. It was further described that keratin phosphorylation was stimulated upon Leukamenin E treatment which was accompanied by “increased soluble granules of keratin” (Fig. 7A, B), but there are no keratin granules shown in this image. Based on the images in Fig. 7A the authors claim that increased soluble pK8-S431 granules induced by Leukamenin E may interfere with initiation of KFs assembly at the cell periphery and block the formation of new KFs network. This statement appears highly speculative.
Author Response
Dear Reviewers,
I sincerely appreciate your professional comments on our manuscript, “Leukamenin E induces K8/18 phosphorylation and blocks the assembly of keratin fiber networks through ERK activation (ijms-716872).” We found these comments very helpful to the revision and improvement of our paper. We have reviewed each one carefully and have revised the manuscript accordingly.
The revised portions are marked in red, and we took this opportunity to make additional improvements not mentioned by the reviewers. We are prepared to make further revisions if necessary. Point-by-point responses to the reviewers’ comments are provided below.
We look forward to your response.
Sincerely,
Responses to the Reviewers’ Comments
Reviewer’s comment: (Given its interference with the activity of major kinases, selectivity for keratins appears highly unlikely. In order to get some solid evidence that the above P-sites are directly responsible for keratin reorganization/solubilization and for the effects studied, the authors need to generate the corresponding P-deficient and P-mimetic mutants and study them in at least one of the cell lines used. This will show to which extent LeuE is selective and to which extent the P-sites identified are responsible for effects in spreading, migration and cell survival. As essential control, knockdown of K8 and K18 has to be performed in at least one cell line. The criterion “solubility” is neither well defined nor investigated.)
Response: Thank you for this suggestion. Indeed, the use of P-deficient and P-mimetic mutants is the most effective method for screening phosphorylation site function.
The assembly-disassembly of keratin filaments is a very complex process, and its mechanism is poorly understood because there are few effective research methods. keratin phosphorylation participates in the regulation of keratin filament assembly and disassembly. Many keratin phosphorylation sites have been identified, but the manner in which they coordinate the assembly-disassembly is unclear. According to current reports, the coordination of multiple phosphorylation sites may be necessary for regulation of keratin filaments assembly-disassembly. We found that leukamenin E could induce phosphorylation of K8/K18 at multiple sites through the ERK pathway, including K8-Ser431, K8- Ser73, and K18-Ser52. There may be other phosphorylated sites at K8/K18 induced by leukamenin E. Elucidating the direct link between these phosphorylation sites and keratin filament depolymerization requires complex and extensive trials. We look forward to further study in the future.
Reviewer’s comment: (It needs to be examined whether LeuE can directly bind to K8 and/or K18 and thereby alter their solubility/biophysical properties. This can be done in vitro using purified keratins)
Response: Thank you for this suggestion. Last year we reported that leukamenin E could covalently modify amino acid residues in a synthetic peptide based on keratin 1 and keratin 10 sequences in vitro by MS analysis (Zhu J et al., 2019). In the study, we found that leukamenin E-induced K8/18 phosphorylation at K8-Ser73/Ser431 and K18-Ser52 also led to inhibition of the assembly of keratin filaments by increased soluble keratin filaments through ERK activation. This suggests that leukamenin E can affect keratin filament assembly through both intracellular and extracellular pathways. Here, we report the regulatory effect of leukamenin E on keratin filaments through the signaling pathway.
Reviewer’s comment: (The fixation conditions described appear strange. Following MeOH fixation (which permeabilizes membranes), detergent extraction is performed. This is likely to affect a number of proteins by solubilizing them. I strongly recommend to omit the latter step and to compare MeOH with formalin/detergens fixation protocols.)
Response:
- Cold methanol fixation protocol from CST's (Cell Signaling Technology, Inc.) instructions for the keratin antibodies. Keratin filaments are very stable. The cells were fixed for only 5 min at negative 20°C. Cellular immunofluorescence images with methanol fixation were clearer than those with formalin fixation.
- Detergent Triton X-100 is a necessary reagent for immunofluorescence production. Its function is to permeabilizs membranes so that antibodies can enter the cell and bind to the antigen.
Reviewer’s comment: (Fig. 1: The corresponding vehicle control is missing. Although the corresponding DMSO concentration is below 0.05 µM one needs to show that there is no DMSO effect (using the highest concentration). Higher concentrations of Leukamenin E lead to the inhibition of proliferation (e.g. 3.5 and 4 µM), which is shown for three cell lines (HUVEC, PLC and PANC-1), Fig. 1B. Apoptosis rates and necrosis seem to be unaffected by Leukamenin E. It is hard to discriminate between early and late apoptotic cells (Fig. 1C) as they all appear yellow. Detailed images would be helpful. In addition, the graphs in Fig. 1D and E need to be changed as the individual bars are not really visible (y-intercept of max. 20%). It appears that 4 µM Leukamenin E increases apoptosis in all three tested cell lines (Fig. 1D). The authors conclude that 2-4 µM Leukamenin E are suitable concentrations for further experiments. However, in Fig. 6B, 7B and 8F even higher concentrations of the compound were used (8 and 12 µM) in HUVEC without testing the toxic effects. Washouts and recovery after 24h are needed.)
Response:
- In this study, we chose the subcytotoxic concentration of leukamenin E. Leukamenin E inhibits cell growth in this concentration range but does not induce apoptosis. In each independent experiment, the concentrations of DMSO in each treatment group were equal. Thank you for this suggestion. Accordingly, we added a DMSO control to Fig. 1B.
- It is essential to show as many cells as possible and a low rate of apoptosis in Fig. 1C, so it is difficult to present the details of the stained cells at any lower magnification (10×10). Considering the reviewer’s suggestion, we added a small window to show different types of cells in the top right corner of Fig. 1C.
- The death rate of HUVECs cells treated with leukamenin E at 12 µM for 12 h was under 6.8% by AO/EB double staining assay. The toxic effects of leukamenin E on HUVECs cells at 12 µM for 12 h is added to text in section 2.1. The reasons for the selection of high concentrations are explained later.
Reviewer’s comment: (Fig. 2: It needs to be better explained why the authors distinguish between spread and spreading cells. In this figure it is hard to believe that depolymerization of MTs and MFs has absolutely no impact on keratin filaments, also because the authors show no co-stainings. In Fig. 2A the Cytochalasin B and Colchicine effect is shown by stainings of the individual cells. For the impact onto keratin filaments completely different cells are shown. In Fig. 2B the corresponding control (DMSO) is missing and again co-staining would be helpful to convince the conclusion.)
Response:
- “Spread cells” and “spreading cells” refer to two different testing systems. In spread cells, keratin filaments are in the dynamic equilibrium of assembly and disassembly. In spreading cells, keratin filaments are undergoing recombination after plating.
- The depolymerization of MTs and MFs has some effect on keratin filaments, manifesting mainly in the recombination of keratin filament network. This does not reduce the density of keratin filaments.
- Cells cultured long-term in vitro are a heterogeneous group of cells, so they show a different tolerance to the drug, which can lead to differences in cytoskeletal changes. We need to identify certain common characteristics of these changes. We focused on the significant decrease in leukamenin E-induced keratin filaments.
Reviewer’s comment: (Fig. 3: The authors claim that in the spreading experiment the assembly of MTs and MFs was inhibited in Cytochalasin B- and Colchicine-containing medium without showing any staining for MTs or MFs. Furthermore it appears that 12h Cytochalasin B and Colchicine treatment affects keratin filament organization (Fig. 3A f and l). So it is not convincing to conclude that the KF network in PLC and HUVEC is independent of MFs and MTs.)
Response:
- For spreading cell systems, we do not show staining for MTs or MFs. The label (data not shown) in our manuscript is missing. This has been revised accordingly.
- As mentioned above, the depolymerization of MTs and MFs resulted in some changes (recombination) in the keratin filament network but did not affect the assembly of the keratin filaments resulting in a decrease in the density of the filament network.
- The depolymerization of microfilaments inhibits the formation of mitotic rings, resulting in the formation of binuclear cells (Fig. 2A, 2B, and 3A). For this reason, the keratin filaments in the cells must be recombined, but the network density did not change. The selection of cytochalasin B and colchicine concentration is crucial.
Reviewer’s comment: (Fig. 4: Again the corresponding DMSO control for the highest concentration has to be shown (Fig. 4A). In Fig. 4B no loading control is shown. Co-staining of the networks as well as quantification would be helpful (e.g. keratin network normal versus bundled for 4 µM compound compared to DMSO).)
Response:
- In the pre-experiment, the cytotoxicity of DMSO and its effect on the cytoskeleton, including microfilaments, microtubules and keratin filaments, were tested to determine the concentration of leukamenin E mother liquor. DMSO had no significant effect on cell growth or the cytoskeleton at 0.5 µM. In each independent cytoskeletal experiment, the concentrations of DMSO in each treatment and control group were equal.
- As mentioned earlier, the cells cultured in vitro are heterogeneous, and the method of co-staining is difficult to truly compare.
Reviewer’s comment: (Fig. 5: The authors claim that in this cell spreading experiment the cells treated with Leukamenin E showed significant inhibition of keratin assembly. The images for the PLCs need to be shown in higher magnification (details in Fig. 5A would be helpful). There could be the possibility that Leukamenin E indirectly alters the ability to spread. But as one can clearly see keratin filaments, it is hard to believe that keratin assembling is inhibited by the compound. In vitro keratin assembly experiments in the presence of Leukamenin E could show a putative inhibitory effect. In addition, biochemical fractionation into soluble and insoluble keratins is needed.)
Response:
- PLC cells spread more slowly than HUVEC cells, so the area of PLC cells was smaller than that of HUVEC cells at the same magnification. Per the reviewer’s suggestion, the PLC cells in Fig. 5 are properly magnified.
- We observed that the reconstruction of the keratin network was delayed but not completely inhibited. This provides a preliminary evidence for further research.
- In addition, biochemical fractionation into soluble and insoluble filaments was carried out using high-speed centrifugation, and quantification of soluble and insoluble keratin was detected by WB assay (Fig. 7B, D).
Reviewer’s comment: (Fig. 6: In this figure Xia et al. investigate, whether Leukamenin E treatment causes keratin K8/K18 phosphorylation by the use of site-specific phospho-antibodies. Fig. 6A and B together confuse the reader and some of the WBs appear over-exposed or the contrast is too high. Again the authors distinguish between well-spread and spreading cells (why?). However, in the spreading cell experiments just HUVECs are shown (Fig. 6B), PLCs not (why?). There is no information in the text, why the authors also used an unspecific pSer antibody in Fig. 6A. The quantification in Fig. 6C and D would be clearer, if the phospho-keratin signal would be normalized against the corresponding unphosphorylated keratin signal (e.g. pK8-Ser431 against K8 signals). The signals of unphosphorylated K8 and K18 or a loading control are even completely missing in Fig. 6B. Again I am wondering, why the authors used so high concentrations (8-12 µM in Fig. 6B)? Vehicle control (especially for the 12 µM treatment) is missing. Based on the WBs in Fig. 6B the authors claim that K8-Ser431 occurred earlier than that of K8-Ser73 and K18-52 in Leukamenin E-treated cells. However, to show a time- and dose-dependency all samples should be run on one gel. Furthermore, it is not clear why there is less phosphorylated K8-Ser73 at 8h (0 and 4 µM) compared to 4h? And why there is also less phosphorylated K18-Ser52 at 4h 8 µM treatment compared to 0, 4 or 12 µM treatment? Quantification in Fig. 6E should be changed like in Fig. 6C and D.)
Response:
- The contrast in Figure 6 has been adjusted.
- “Spread cells” and “spreading cells” refer to two different testing systems. We can observe leukamenin E-induced alteration in keratin filaments in two systems.
- An unspecific pSer antibody was used to determine whether Leukamenin E had a phosphorylated activation at serine sites in the cells (Fig. 6A).
- We have incorporated the suggested changes into Fig. 6B, C, and D.
- Leukamenin E-induced phosphorylation at K8-Ser73 and K18-Ser52 requires at least 16 h to spread after plating at 4 µM. We used high concentrations in spreading experiment. Leukamenin E-induced stimulation (Ser52 and Ser73) at high concentrations (8–12 µM) takes only 8 h.
- Concerning “K8-Ser73 at 8 h (0 and 4 µM) compared to 4 h,” the contrast in the figure has been adjusted.
- Concerning “there is also less phosphorylated K18-Ser52 at 4 h 8 µM treatment compared to 0,” we consider that sample (pK18-Ser52 at 4 h 8 µM) may have been subject to overflow error in electrophoresis experiment.
- The quantification in Fig. 6E has been changed.
Reviewer’s comment: (Fig. 7: The authors performed double-stainings using K18 together with K8-Ser431, -Ser73 or K18-Ser52 antibodies to show changes in the phospho-K8/K18 signals after Leukamenin E treatment. The described changes are hard to see; detailed and confocal images would be helpful. As the bundled keratin filaments appear brighter (Fig. 7A h), it could be that the filaments at the cell periphery appear darker. Furthermore, the control shows also a cell with keratin bundling around the nucleus (Fig. 7A m). In Fig. 7B the vehicle control is missing as well as a loading control or Ponceau S staining. How the two keratin pools were made (supernatant, precipitate); there is no information in the methods sections. Why the authors used a Pan-Keratin antibody in Fig. 7B and no K8 or K18 antibody? This signal could be better compared with the phospho-signals of the WBs from Fig. 7D. The quantification in Fig. 7C does not reflect the WBs in B, e.g. compare supernatant signals of 0 and 1/4 µM of well-spread to spreading cells. The signal for the spreading cells at 4 µM appears higher compared to 1 µM in well-spread cells, but the quantification shows nearly equal relative intensities? Also in Fig. 7D a loading control is missing and some of the blots appear over-exposed.)
Response:
- We used the same parameters to take double-staining pictures under a microscope. We are convinced that the observations are reliable. It is important that the results of the double-staining are consistent with those of the WB analysis (Fig. 7D). Considering the reviewer's suggestion, the double-staining pictures (Fig. 7A) have now been properly magnified using Photoshop software.
- The preparation method of supernatant and precipitation was added to the experimental method section (4.5 Western blot).
- K8 /18 antibodies are more suitable for this experiment than pan-keratin antibody. The epithelial cells mainly express K8/18, so pan-keratin largely reflects the content of K8/18.
- The over-exposed blots in Fig. 7B (spread cells) resulted in high relative intensity (Fig. 7C, spread cells). So, we adjusted the contrast of the picture, re-measured the gray value, and modified the graph (Fig. 7C, spread cells).
- Pre-experiments showed that 0.5 µM of DMSO had little effect on the expression of keratin and its keratin filament network.
Reviewer’s comment: (Fig 8: The authors next determined whether ERK is involved in Leukamenin E-induced K8/K18 phosphorylation as it is already known that this kinase can phosphorylate K8. They show that the compound stimulated ERK phosphorylation in a dose-dependent manner. Here I would also normalize the phospho-signals against unphosphorylated protein signals (Fig. 8B and C). Fig. 8D shows no unphosphorylated K8/K18 or a loading control. It is also necessary to show that U0126 really blocks MAPK pathway (e.g. phosphorylation of downstream p38) under this conditions. Using PLCs: there is no effect onto pK8-Ser73 and pK18-Ser52 upon ERK inhibitor use. Furthermore, there is no text information in the manuscript about Fig. 8E-G?)
Response: We apologize for this error and have made the necessary corrections.
- The unphosphorylated K8 and K18 have been added to Fig. 8D.
- P38 and ERK are two parallel signaling pathways in MAPK pathway. P38 Inhibitors do not affect Leukamenin E-induced K8/K18 phosphorylation. The data are not shown in the manuscript.
- We replaced the image in Fig. 8D for PLC cells.
- The information about Fig. 8E–G has been added to the text in the manuscript.
Reviewer’s comment: (Fig. 9: The authors next investigated the impact of Leukamenin E onto cell migration. Using a transwell as well as a wound healing assay they claim that the compound induced an inhibitory effect on migration of HUVEC and Panc-1 cells. It was shown by others that small compounds like TPA or SPC can promote migration of Panc-1 cells. This could be a good positive control and added to the experiments. Also due to the fact, that proliferation rates were slightly decreased (Fig. 1B, compare control to 4 µM Leukamenin B treatment of Panc-1 cells) the inhibitory migration effect could be due to proliferation effects. To avoid this, a proliferation inhibitor could be also used in the wound healing assay. Minor point: show the gap closure as long as the control is completely closed (longer time points)
Response: In our experiment, a proliferation inhibitor, mitomycin-C (final concentration: 10 mg/mL) was added to the plates to inhibit cell proliferation. Please refer to lines 545-546 of the manuscript.
Reviewer’s comment: (Discussion: It is not convincingly shown that Leukamenin E exerts its effects on keratin filaments independently of other major cytoskeletal components. The authors propose that Leukamenin E can specifically inhibit the assembly of KFs, however a lot of KFs are seen in all the images after compound treatment. The statement “Leukamenin E presumably hinders KFs assembly in vivo because it can covalently link to synthetic peptides in vitro” needs to be better explained. It was further described that keratin phosphorylation was stimulated upon Leukamenin E treatment which was accompanied by “increased soluble granules of keratin” (Fig. 7A, B), but there are no keratin granules shown in this image. Based on the images in Fig. 7A the authors claim that increased soluble pK8-S431 granules induced by Leukamenin E may interfere with initiation of KFs assembly at the cell periphery and block the formation of new KFs network. This statement appears highly speculative.)
Response:
- In cells where microfilament and microtubules are depolymerized, the keratin filaments networks remain intact, suggesting that they are independent of microfilament microtubules. The term “independence” is relative rather than absolute. Studies have shown that there are connections between microfilaments, microtubules, and intermediate filaments in cells, but in epithelial-type cells, keratin filament networks are relatively independent. We can use models of epithelial cells to screen compounds that affect the assembl y of keratin fibers (Zhu et al., 2019).
- Keratin filaments are very stable and their turnover in the cells is slow. No compounds have been found for rapid depolymerization of keratin filaments. In our study, Leukamenin E resulted in a significant reduction in keratin filaments in the spread cells and inhibition of reconstitution of keratin filaments in the spreading cells.
- The conclusion “Leukamenin E presumably hinders filaments assembly in vivo because it can covalently link to synthetic peptides in vitro” is from our previous report, please refer to Ref. Zhu et al. (2019).
- In cells, the assembly or disassembly of keratin filaments is graded: It includes dimers, tetramers, and other polymers. These different levels of polymer are called keratin particles (granules). These nanoscale particles are difficult to observe under an optical microscope. In Fig. 7A, leukamenin E resulted in a weakening of the fluorescence staining of the keratin filament network, suggesting that some depolymerized keratin filament particles entered the cytoplasmic matrix. In Fig. 7B, the increase of keratin in the supernatant of the cell lysate and the decrease of keratin in the precipitate established that the particles depolymerized from the keratin filament network entered the cytoplasmic matrix.
Round 2
Reviewer 1 Report
The authors have answered to my criticisms.
Reviewer 2 Report
Except one experiment in which the effect of LE on keratin solubility was examined, no experiemnts were performed to strengthen the claims made in the original ms.
1. The central hypothesis is that LE increases keratin solubility by increasing phosphorylation at K8-Ser73/431 and at K18-Ser via the ERK pathway. This needs to be examined by a real experiment, i.e. in vitro mutagenesis of these and possibly other sites, followed by transfection studies. None of the arguments put forward by the authors can replace such an experiment.
2. The protocol used to investigate keratin solubility following LE treatment is not helpful. In the absence of information on buffer composition (detergent, ionic strength, pH), "solubility" remains an undefined term.
Author Responses
Reviewer’s comment: (Except one experiment in which the effect of LE on keratin solubility was examined, no experiemnts were performed to strengthen the claims made in the original ms.
1. The central hypothesis is that LE increases keratin solubility by increasing phosphorylation at K8-Ser73/431 and at K18-Ser via the ERK pathway. This needs to be examined by a real experiment, i.e. in vitro mutagenesis of these and possibly other sites, followed by transfection studies. None of the arguments put forward by the authors can replace such an experiment.)
Response: It has been shown in previous works that the phosphorylation of keratin affects its solubility. It has been known for a long time that the equilibrium between the soluble and filamentous keratin pool is determined by phosphorylation (Feng et al., 1999; Strnad et al., 2002; Woll et al., 2007; Zhou et al., 1999). Keratin phosphorylation likely contributes to regulation of the switch between the filamentous and soluble keratin pool (Sawant M, 2017). Busch T, et al. (2012) generated eCFP-tagged mutants of K8 and eYFP-tagged mutants of K18 that mimic phosphorylation at K8(S431) and K18(S52), respectively, or exhibit a non phosphorylatable site. And they observed that there was more K8(S431E, mimicking phosphorylation) detectable in the cytosolic/soluble fraction than K8(S431A, not phosphorylated) and K8 WT. The transfection of K8(S431E) alone was also sufficient to induce perinuclear organization of K8 and K18 in human pancreatic and gastric cancer cells. These findings suggested that the phosphorylation at K8(S431) induced perinuclear organization of endogenous K8 and K18 by increasing the soluble fraction of K8/K18 filaments (Busch T, et al., 2012). This experiment confirmed that phosphorylation of K8-Ser431 sites could increase the solubility of keratin filaments (KFs).
In our paper, we observed that leukamenin E induces KF network destruction in well-spread cells and inhibition of KF network assembly in spreading cells, largely independent of microfilaments and microtubules, and accompanied by keratin phosphorylation at K8-Ser73/Ser431 and K18-Ser52, and by increased soluble fraction of KFs as well as phosphorylated K8-Ser73/Ser431 and K18-Ser52 in PLC and HUVECs.
The selective MEK inhibitors U0126 not only inhibited leukamenin E-induced phosphorylation at the three sites mentioned above but also increased soluble fraction of KFs. This indicated a close or direct relationship between leukamenin E-induced phosphorylation at K8-Ser73/Ser431 and K18-Ser52 and increased soluble fraction of KFs. This supports our conclusion that leukamenin E increases KF solubility by increasing phosphorylation at K8-Ser73/431 and at K18-Ser via the ERK pathway. The use of inhibitors to establish the downstream relationship of proteins in signaling pathways is currently a common and widely accepted experimental method.
However, we know that the mutation experiment suggested by the reviewer is the most effective method at present to demonstrate the specific effects of the phosphorylation at K8-Ser73/Ser431 and K18-Ser52 on increased soluble fraction of KFs. We are also aware that the description of the specific effects of leukamenin E is inaccurate. Its specificity needs further verification by using the mutation experiment suggested by the reviewer. However, given our current situation, we cannot complete the mutation experiment in a short time because we don't have an experimental platform to generate mutants. Based on current experimental evidence, we revise our conclusion on the specific role of leukamenin E, and clearly stated that leukamenin E may also induce phosphorylation at other sites on K8/K18 leading to increased soluble fraction of KFs in this paper.
Reviewer’s comment: (The protocol used to investigate keratin solubility following LE treatment is not helpful. In the absence of information on buffer composition (detergent, ionic strength, pH), "solubility" remains an undefined term.)
Response: The protocol used to investigate keratin solubility were performed with slight modifications according to the method of Beil M, et al. (2003). The main components of the lysis buffer, including detergent, ionic strength, protease and phosphatase inhibitors, have been listed in the experimental section of our paper. We are very sorry that we neglected to describe the details of the experimental method in previous manuscript.
In much of the literature on keratin phosphorylation-induced KF solubility, researchers have used a similar protocol for qualitative and quantitative analysis of soluble or insoluble keratin. These experimental methods, including centrifugation combined with WB assay, immunofluorescence, or electron microscopy, are widely used by researchers. Immunofluorescence and centrifugation combined with WB assay were used in our experiments.
References
Beil M, Micoulet A, Wichert G, Paschke S, Walther P, Omary M, Veldhoven P, Gern U, Wolff-Hieber E, Eggermann J, Waltenberger J, Adler G, Spatz J, Seufferlein T, Sphingosylphosphorylcholine regulates keratin network architecture and visco-elastic properties of human cancer cells, Nature Cell Biology 5 (2003) 803-811.